# GLOBAL-FATE (version 1.0.0): A GIS-based model for assessing contaminants fate in the global river network

Carme Font[1,2], Francesco Bregoli[1, *], Vicenç Acuña[1,2], Sergi Sabater[1, 3], Rafael Marcé[1,2]

[1] Catalan Institute for Water Research (ICRA), Emili Grahit 101, 17003 Girona, Spain
[2] University of Girona, Girona, Spain
[3] Institute of Aquatic Ecology, University of Girona, Campus Montilivi, 17071 Girona, Spain
[*] Current address: Water Science and Engineering Department, IHE Delft Institute for Water Education, Westvest 7, 2611 AX, Delf, The Netherlands

*Correspondence to*: Rafael Marcé (rmarce@icra.cat)

**Abstract.** GLOBAL-FATE is the first open-source, multi-platform, user-friendly, and modular contaminant fate model operating at the global scale linking human consumption of pharmaceutical-like compounds with their concentration in the river network. GLOBAL-FATE simulates human consumption and excretion of pharmaceuticals, the attenuation of the contaminant load in wastewater treatment plants, as well the attenuation of the contaminant load in river reaches, lakes, and reservoirs, as a first order decay depending on residence time. We provide a comprehensive description of model equations and the overall structure of the model, with spacial attention to input/output datasets. GLOBAL-FATE is written in C, can be compiled in any platform, and uses inputs in standard GIS format. Additionally, the model can be run inside QGIS as a plug-in. The model has no built-in working resolution, which depends on the intended use and the availability of appropriate model inputs and observed data. We exemplify the application of GLOBAL-FATE solving the global concentration of diclofenac in the river network. A comparison with a dataset of diclofenac concentration observations in rivers suggest that GLOBAL-FATE can be successfully applied in real case modelling exercises. The model is particularly sensitive to the generation of contaminant loads by human pharmaceutical consumption, and to the processes governing contaminant attenuation in the river network. GLOBAL-FATE will be a valuable tool for the scientific community and the policymaking arena, and could be used to test the effectiveness of large scale management strategies related to pharmaceutical consumption control and wastewater treatment implementation and upgrading.

## 1 Introduction

The United Nations 2030 Agenda for Sustainable Development identifies 17 master goals, amongst which is the availability and sustainable management of water and sanitation. This agenda establishes as a goal the improvement of water quality by reducing pollution, eliminating dumping, and minimizing release of hazardous chemicals to the river network by 2030 (UN, 2015). However, a large proportion of surface water networks is currently severely affected by sewage inputs from waste

water treatment plants (WWTP) or by direct sewage disposal (Richardson et al., 2005; Stewart et al., 2014; Hernández et al., 2015; K'oreje et al., 2016). Sewage disposal inputs organic matter, nutrients, and fecal bacteria to the river network, together with a whole plethora of chemicals related to household human activities. These include micro-plastics and nanomaterials (Besseling et al., 2017), pharmaceuticals (Li et al., 2016), personal care products (Arlos et al., 2014), and even illicit drugs

(Postigo et al., 2010). This increment of down-the-drain chemicals reaches to the river network and affect both humans and biodiversity and ecosystem function (Rudd, 1970), so forth posing at risk water security (Goldman and Koduru, 2000; Vörösmarty et al., 2010).

Assessing chemical discharges and their fate in the river network is thus vital to evaluate both the health of aquatic ecosystems and the security of water supplies for human needs. This requires adequate models to consider the spread and

dynamics of chemicals at large spatial scales, both for assessing the current water quality status in regions with poor monitoring programs coverage (Strokal et al., 2019), and for planning management and mitigation measures. Existing models approach the fate of contaminants in multimedia (air, water, soil) and using steady-state models working at regional scales such as ChemCAN, HAZCHEM, or EUSES (MacLeod et al., 2011; Gouin et al., 2013; Lindim et al., 2016). Others are process based, operating at the watershed scale, and perform as dynamical in-stream water quality models, such as

MIKE11, SWAT, WASP, QUAL2E, or DELWAQ (Liang et al., 2015; Santhi et al., 2005; Di Toro et al., 1983; Brown et al., 1987; Van Wijngaarden, 1999). Another set of models analyse the dynamics of down-the-drain pollutants, considering the linkages between engineered systems (e.g., WWTP) and natural systems (e.g., rivers). This includes PhATE, GREAT-ER, LF2000-WQX, STREAM-EU, iSTREEM, or ePiE (Anderson et al., 2004; Feijtel et al., 1997; Johnson et al. 2007; Boxall et al., 2014; Lindim et al., 2016; Kapo et al., 2016, Oldenkamp et al., 2018), and frequently target on pharmaceutical products

though are also suitable to simulate the fate of any compound decreasing following first order decay dynamics (Table 1). Particularly, models in Table 1 are applied for chemicals whose dominant emission source to the environment is via WWTP effluents. Most of these models are highly data demanding and use many adjustable parameters. This makes some of them computationally inefficient; others have non-open source codes, which make their use for global or continental scale calculations cumbersome.

Recently, other approaches specifically designed for very large scales have used a Geographical Information System (GIS) framework to solve the routing of chemicals along the river network (Pistocchi et al., 2012; Dumont et al., 2015; Grill et al., 2016; Rice and Westerhoff 2017). Most of these models use a much simpler model parameterization, in order to make continental and global calculations accessible. However, some of them assume that chemicals do not decay when travelling through the river network, and simply rely on dilution factors once pollutants enter in the river network. Further, they work

at a fixed spatial scale which is either very rough to adequately represent the river network (e.g., 0.5 degrees), or too detailed to be practical for global calculations due to computational requirements (e.g., 500 m, Grill et al., 2018).

GLOBAL-FATE has been designed to overcome these constraints, offering the first contaminant fate model operating at the global river network, including lakes and reservoirs, which is at the same time open-source, multi-platform, user-friendly, and modular. This will make global contaminant calculations accessible to a much wider community of scientists and practitioners, opening the door for including pharmaceutical pollution into influential assessments of climate change impacts (e.g., the Inter Sectoral Impact Model Intercomparison project) and global policy instruments like the UN Sustainable Development Goals agenda. GLOBAL-FATE calculates the steady-state concentration of a user-defined down-the-drain contaminant through the global river network, including lakes and reservoirs. GLOBAL-FATE is offered as an open-source, GIS-based model programmed in the C language, allowing researchers to select the input information (water routing, hydrology, population, etc.) and the spatial resolution at which the model has to perform. So forth, the model can include new or different hydrological datasets and other input information, and hence it is not fundamentally restricted to a single modelling resolution, hydrological, or socio-economic scenario. The model simulates the propagation of down-the-drain contaminants along the river network, and the constituent decreases at a rate proportional to its concentration in the aquatic media. GLOBAL-FATE is also computationally efficient, can be run in Windows or Linux machines, and can take advantage of parallel computing in multi-processor computers or clusters. It can also be run as a user-friendly plug-in in QGIS, and the modular structure of its code allows switching different functions of the model on and off. Here we describe the structure, functioning, and strengths and limitations of GLOBAL-FATE. First, we explain the structure and functioning of the model, focusing on the type of input data structure and the formulation of the different hydrological and biogeochemical processes. Then, the application of the model is exemplified solving the worldwide propagation of the pharmaceutical diclofenac throughout the global river network. Finally, we discuss strengths and limitations of GLOBAL-FATE, and point at future developments.

## 2 Methodology

GLOBAL-FATE is a physically-based model for simulating constituent inputs to the river network and their routing along the river network at the global scale. Our approach shares key assumptions and modelling mechanisms with other large scale pharmaceutical models for the river network (i.e., Keller et al. 2006; Pistocchi 2014; Grill et al., 2019), including the use of per capita mass emissions of the contaminant of interest, simplified parameterization of losses due to human metabolism and removal in wastewater treatment plants, and dilution and first order attenuation dynamics upon discharge into natural waters. However, GLOBAL-FATE is the first model natively operating at the global scale including all those mechanisms, including explicit routing and attenuation in lakes and reservoirs.

The model uses GIS input files to solve for the contaminant fate at every element (cell) of the domain (raster). The model is multi-platform, written in C, and uses multi-core parallel computing via OpenMP. Additionally, GLOBAL-FATE can be used from QGIS (QGIS Development Team, 2018) as a plug-in, so it can be executed in a *push-button* fashion, in order to load all

the layers and information the model needs in a user-friendly way, as well as automatically producing basic visualizations of the results. The code (including compilation instructions) is freely available at https://github.com/ICRA/GLOBALFATE,

including the QGIS plug-ín. Pre-built executables are available under request.

GLOBAL-FATE simulates the fate of contaminants that behave as human pharmaceuticals (Fig. 1). That is, the model assumes that the origin of the contaminant load is the consumption of a pharmaceutical by population, which can differ in different regions of the World considering population density and per capita consumption. No other origins, such as diffuse sources through spreading of pharmaceutical-rich farm manure on agricultural fields, are currently included. The model

assumes an excretion rate by population, and the fraction finally excreted will reach the river network either directly or after some attenuation in wastewater treatment plants (WWTPs). The fraction of load treated can be dependent on the region of the World, while the decay in WWTPs is an input parameter that applies globally. Finally, the contaminant load is routed along the river network, considering that the contaminant will decay following first-order kinetics dependent on water residence time in the river network reaches. GLOBAL-FATE also considers the presence of lakes and reservoirs in the river

network and includes a particular solution for water residence time in these systems in order to calculate contaminant decay. The main output is a global map of predicted contaminant load or concentration throughout the river network.

The model workflow (Fig. 2) is based on the input of 9 global maps in the form of raster datasets and the definition of 8 parameters (Table 2). The model has been designed to work with raster data with the geographic coordinate system WGS84 in decimal degrees, but it has not a predefined spatial resolution. The only prerequisite is that all input rasters must have the

same resolution and extent. Raster input data files are expected to have ASCII ESRI grid format, which makes GLOBAL-FATE very easy to set up using customary GIS software even for non-experienced users.

## 2.1 Model workflow

The model is composed of six main functions that are executed successively, being some outputs the inputs to the next

function. First, the geographical related functions are run, which deliver streamflow and water residence time along the river network as main outputs. Then, the contaminant related functions calculate the load of contaminant to the river network by population, after discounting for wastewater treatment, and also the routing of the contaminant through the river network considering that the contaminant decays following a first order reaction dynamics. The following is a description of the calculations performed by the six main functions in GLOBAL-FATE (see also Table 3 and the code and example input files

at the GitHub repository).

### 2.1.1 Cells area (function *Area_m2_fun.c*)

This function performs an auxiliary calculation for the flow routing function. In order to route locally generated runoff the area (m²) of each cell in the domain is necessary. Considering WGS84 as a reference coordinate system, cells height is the length of the arc formed by the angle $\delta$ (raster resolution in decimal degrees transformed to radians) and is constant in the whole grid, calculated as:

$$H = \delta R, \tag{1}$$

where R is the authalic Earth radius ( $6{,}371{,}007.2$ m). In its turn, the width of each cell depends on latitude:

$$W(y) = R\left(\sin\left(y + \frac{\delta}{2}\right) - \sin\left(y - \frac{\delta}{2}\right)\right), \tag{2}$$

Here $y$ corresponds to the latitude and comes from:

$$y = y_0 + \delta(nr - j), j = 1, \dots, nr, \tag{3}$$

where $y_0$ is the southern cell latitude and $nr$ is the number of rows in the raster, so *j* is a latitude index. Due to the fact that cells at the same latitude have equal width, both area (m²) and width are calculated for one meridian:

$$A(y) = H \cdot W(y), \tag{4}$$

### 2.1.2 Flow routing (function *Flow_accumulation_m2.c*)

Streamflow in each cell of the raster is computed using a runoff accumulation approach. First, for each basin, cells are enumerated from headwaters ( $l=0$ ) to river mouth ( $l=L$ ), following the hierarchical organization of the river network. The $J_l$ is the set of cells indexes at stage $l=0,\dots,L$ .. This cells classification is obtained as a raster input dataset (Table 3), and is a typical product of many GIS hydrological algorithms (often defined as an area or flow accumulation layer). Second, we define the amount of runoff locally generated in each cell due to the precipitation-evaporation water budget. For this, we use available products of mean annual runoff (m year⁻¹) at the global scale in raster format, re-scaled to the same resolution as the rest of the hydrological input rasters. Finally, we route the locally generated runoff along the river network following the hierarchical order defined in the first step. Streamflow in cell $j$ in $J_{l>0}$

is computed as the sum of runoff inputs from the surrounding cells and that generated within the cell. In order to determine the input from the neighboring cells, a raster of flow direction is used. Such raster must be encoded following the D8 method (O'Callaghan et al., 1984). Finally, average annual streamflow in $m^3 \, year^{-1}$ in each cell is calculated after summing up the multiplication of cells runoff ($m \, year^{-1}$) by the corresponding cells area (Eq. 4, $m^2$):

$$q_j = \sum_{i \in N_j} q_i + runoff_j A_j, j \in J_{l>0}, \quad (5)$$

where $N_j \subseteq J_{l-1}$ is the set of indexes of the neighborhood of cell $j$, such that $i \in N_j$ implies that flow of cell $i$ goes to cell $j$, and $A_j$ is the area of cell $j$. Note that for $l=0$, $J_0$ represents the set of headwater cells, where there is no neighboring inputs, and Eq. 5 simplifies to:

$$q_j = runoff_j A_j, j \in J_{l=0}, \quad (6)$$

### 2.1.3 Water residence time calculation in river cells (function *RT_rivers_calculator.c*)

Residence time (RT) of water in rivers is a key magnitude for the calculation of contaminant decay in the river network. RT at each cell is calculated as the division of the longitude (m) of the river reach (cell) by water velocity ($m \, s^{-1}$), i.e.;

$$RT = \frac{l}{v}, \quad (7)$$

The longitude of the flow path through the cell depends on its direction. We differentiate four cases:

$$l = \begin{cases} H, \text{if the flow has North or South direction} \\ W, \text{if the flow has East or West direction} \\ \sqrt{H^2 + W^2}, \text{if the flow has NE, NW, SE, or SW direction} \end{cases}, \quad (8)$$

$H$ and $W$ correspond to cell height and width explained in section 2.2.1. The flow velocity within the cell is calculated using the Manning equation:

$$v = \frac{1}{n} R_h^{2/3} S^{1/2}, \quad (9)$$

$$R_h = \frac{wh}{2h+w}, \quad (10)$$

where $n$ is the Manning coefficient (s m$^{-1/3}$), $R_h$ is the hydraulic radius (m) and $S$ is the local slope (m m$^{-1}$), obtained as an external input in the form of a slope raster dataset. The Manning coefficient is applied globally, in our case we chose 0.044 s·m$^{-1/3}$ following Schulze et al. (2005). The hydraulic radius is calculated after solving for channel width ($w$) and depth ($h$) using the power functions of Leopold and Maddock (1953):

$$w = a\, q^b, h = c\, q^d \quad , \tag{11}$$

where $a, b, c$ and $d$ are fitted parameters (in our case we chose $a = 7.2, b = 0.5, c = 0.27, d = 0.39$ after Andreadis et al., 2013), and $q$ is river discharge (m$^3$ year$^{-1}$).

### 2.1.4 Water residence time calculation in lakes and reservoirs (function *RT_lakes_incorporation.c*)

Lake and reservoirs are included in GLOBAL-FATE using available global databases on the location, shape, and volume of lakes and reservoirs. These spatially explicit databases must be converted into a raster with the same resolution and projection as the other hydrological rasters. The general strategy is to store all features of a given lake (volume, residence time) in the outlet cell (i.e., the cell routing the streamflow downstream from the lake), making the rest of cells of the lake as mere pipes of water and constituents to that outlet cell, where all contaminant reactions occur. Since most lakes occupy more than one cell in the network, the indexes of the cells belonging to a lake (raster of lakes location and shape) need to be indicated. Being $L_j$ the set of indexes of the cells belonging to lake $j$, streamflow to lake as calculated by Eqs. 5 and 6, $Q_j$, corresponds to the outlet cell, i.e., the cell with maximum flow accumulation:

$$Q_j = max\left[ q_i, i \in L_j \right], \tag{12}$$

And the RT for the lake is the quotient of its volume, $V$, and streamflow (m$^3$ year$^{-1}$):

$$RT = \frac{V}{Q}, \tag{13}$$

The volume of the water bodies, $V$ (m$^3$), is introduced as a raster input dataset (Table 3) in which the volume information for a particular lake is stored in the outlet cell. This implies that during RT calculation for lakes and reservoirs the cell corresponding to the lake outlet will store the annual average residence time value for the entire lake, while the rest of cells of the lake will be considered as dummy cells in terms of residence time. In its turn, this implies that during calculation of

contaminant transport and reaction throughout the network, only the outlet cell of a lake will be reactive in terms of contaminant decay. Thus, the rest of cells pertaining to that lake will transport water and constituents, but all contaminant decay will take place exclusively in the outlet cell. The final implication is that lakes and reservoirs are treated in GLOBAL-FATE as point-like features, with no spatial heterogeneity. The RT raster for the river network obtained using Eq. 7 is finally updated with the RT for lakes and reservoirs (RT for the entire lake in the outlet cells, and a dummy RT value (-9999) for the rest of lake cells).

### 2.1.5 Contaminant load to the river network (function *Initial_contaminant_load.c*)

The contaminant load to the river network in GLOBAL-FATE is modelled for a constituent that behaves as a human pharmaceutical. Consequently, load from each cell in the raster domain is modelled as a function of the population present in each cell and several parameters accounting for consumption and excretion by population, and contaminant decay in wastewater treatment plants (WWTPs) before the contaminant mass is loaded into the river network. The contaminant load to the river network ($L_0$) is thus defined as:

$$L_{0,j} = \gamma\, m_j P_j \left(1 - w_{treat}\, \varepsilon \right), \tag{14}$$

where $j$ is the cell index, P is the population raster, $m$ is the compound per capita consumption raster (g person$^{-1}$ year$^{-1}$), usually defined at the country level), and $\gamma$ is a parameter for the human excretion rate. The second term in the equation expresses the loss of contaminant due to wastewater treatment, and includes the proportion of population that is connected to WWTPs ($w_{treat}$ usually available at the country level), and contaminant removal rate during wastewater treatment ($\varepsilon$), which needs to be calibrated or assigned to bibliographical values. The output of Eq. 14 is the contaminant load (g year$^{-1}$) discharged by any populated cell; this amount is used as initial values in the contaminant routing function.

### 2.1.6 Contaminant routing (function *Contaminant_accumulation.c*)

The contaminant routing along the river network assumes that once delivered to the river network the contaminant load decays following a first order reaction kinetics:

$$\frac{dC}{dt} = -kC \; , \tag{15}$$

where $k$ is the first-order decay constant (hour$^{-1}$). After reaction during a given period of time, the remaining load will be defined by the solution of the differential in Eq. 15:

$$C(t) = C_0 e^{-kt},$$ (16)

where time $t$ would correspond in GLOBAL-FATE to the time (hours) that the constituent remains into the cell, i.e., the water residence time ($RT$) previously calculated with Eqs. 7 or 13. However, to solve the routing of the contaminant along the network, we also have to take into account the hierarchical relationship between cells. In computational terms, this
function works similarly to the flow routing function, with the difference that we have to implement not only the transport of the contaminant, but also the decay in Eq. 16. In this context, the load of contaminant in a cell $j$ considering loading from upstream cells and its own local population and first order decay in the cell is defined by:

$$L_j = \left( \sum_{i \in N_j} L_i + L_{0,j} \right) e^{-k\,RT_j}, \; j \in J_{l>0}, N_j \subseteq J_{l-1},$$ (17)

where $L_i$ is the load from upstream cell $i$, $L_{0,j}$ is the load from local sources (Eq. 14) in cell $j$, and $RT_j$ is residence time in cell $j$. From this load we can calculate the resulting contaminant concentration in cell $j$ ($C_j$, g m$^{-3}$) with:

$$C_j = \frac{L_j}{q_j},$$ (18)

where $q_j$ is streamflow in cell $j$. Considering that we have both transport and a first order decay process, the contaminant routing must be solved respecting the hierarchical arrangement of the river network, that is, all contributing upstream cells must be calculated before a particular cell can be solved.

## 2.2 Coding general strategy

GLOBAL-FATE has been programmed in C. C is a compiled language, so it implements algorithms and data structures
swiftly, facilitating faster computation. Furthermore, the use of loops is not as punishing as in interpreted languages, such as Python, R, Matlab, or Octave, which is relevant in a code that has loop structures to solve the water and contaminant routing. Regarding this, we integrated parallelization routines in the code using OpenMP to expedite calculations during time-consuming loop calculations and raster input/output routines. OpenMP supports multi-platform shared memory multiprocessing programming in C. It works out well for any multi-core machine, while still executable in single core
computers. The model has been coded using a modular structure in several independent functions, so it is possible to skip the hydrological calculations if they are not relevant for a given analysis (for instance, different wastewater treatment scenarios can be solved without running the hydrological functions every time), but we also offer the possibility to trigger the whole model chain in a single call. The model has been also designed to take command line arguments when executed, if necessary.

This enables the use of pseudo-parallelization to run different model instances with different input arguments, for instance to perform automatic calibration or sensitivity analyses.

Some readers might be surprised by the fact that we programmed our own flow routing function instead of using a customary flow accumulation algorithm from one of the multiple hydrological GIS packages available. This stems from the fact that the contaminant routing function cannot be solved with a standard flow accumulation algorithm with a "negative weights" raster to solve for contaminant decay. The hierarchical nature of the river network is intimately related to contaminant transport and decay, and the process in non-linearly dependent on the mass present in each cell, so there is no way of defining *a priori* a "weighting" raster to solve contaminant transport with a standard flow accumulation algorithm. This means that we had to code the accumulation and decay of the contaminant so that contaminant mass is calculated in each cell appropriately. It is easy to realize that setting the first order decay constant to zero in our code gives a solution that would be similar to the one delivered by a standard flow accumulation algorithm. We decided to calculate flow routing with our algorithm to avoid using two different codes for flow routing and contaminant transport. Although both algorithms would use the same flow direction raster and thus should produce coherent results even using two different codes, we preferred to ensure a total coherence between the two solutions (water and contaminant). Moreover, the fact that our code is programmed in a compiled language with OpenMP parallelisation for loops makes our flow routing algorithm as efficient as any customary GIS flow accumulation function.

## 3. Example model application: concentration of diclofenac in the global river network

Here we exemplify the application of GLOBAL-FATE, simulating the concentration of diclofenac in the river network at the global scale. Diclofenac is a non-steroidal anti-inflammatory drug used as an analgesic, anti-inflammatory and antipyretic for humans (Todd and Sorkin, 1988). Diclofenac enters the environment through treated or non-treated wastewater discharges (Pistocchi et al., 2012) and it has been shown affecting aquatic organisms (Nassef et al., 2010). Furthermore, this pharmaceutical was included in the EU watch list of emerging contaminants of the Water Framework Directive by the European Commission (EC) under the Water Framework Directive (WFD), as well as by the US Environment Protection Agency (US EPA), with a proposed maximum acceptable concentration of 100 ng L$^{-1}$ (Acuña et al., 2015).

### 3.1 The input data sets

All rasters in this example were re-scaled and adjusted to match a resolution of 1/16 deg ( $\delta$ ), with extreme positions $x_0 = -180$ (western cell position) and $y_0 = -56$ (southern cell position), and for extension $nr = 2240$ (number of rows) and $nc = 5760$ (number of columns). We want to stress here that the following collection of datasets is just one possible choice; GLOBAL-FATE is not restricted to work with those datasets or resolutions. Researchers are free

to choose the data products that best serve the interest of the research question at place. All the example datasets are available in the GitHub repository, and correspond to the datasets identified in Table 3.

### 3.1.1 Morphology and Hydrology

*Flow direction and area accumulation rasters*. We used the Dominant River Tracing (DRT) (Wu et al., 2012), a database designed to perform macro scale hydrologic calculations, to build the global river network. We used the flow direction raster at 1/16 of a degree (approx. 7 km) in http://files.ntsg.umt.edu/data/DRT/ to generate a hierarchical cells order raster using the area accumulation algorithm in ESRI ArcGIS Spatial Analyst.

*Runoff raster*. As a runoff raster, we used the composite global annual runoff from Fekete et al. (2002), which consists in a raster of annual runoff with values in mm year$^{-1}$. The original raster was rescaled to the same resolution and extent as the other hydrological raster, disaggregating the runoff raster so that the water mass remained the same after disaggregation.

*Slope raster*. The slope raster was produced in QGIS from the digital elevation models at approximately 1 km resolution in HydroSHEDS (http://hydrosheds.cr.usgs.gov) and Hydro1k (USGS, https://lta.cr.usgs.gov/HYDRO1K) for regions above 60 N.

*Lakes locations and shape raster*. To identify the location and shape of lakes and reservoirs we merged the GRanD database for reservoirs (Lehner et al., 2011) with GLWD (Level 1) for lakes (Lehner & Doll 2004). Duplicate lakes were removed before producing the final map.

*Lakes volume raster*. To produce the volume raster, we first identified the pixel with the largest streamflow for each lake and reservoir, and then we stored the volume information for each lake in that particular pixel. The volume of the 41 World biggest lakes was manually introduced after literature review. For reservoirs, the GRanD database already contains the volume of each system, while for lakes volume is not available for all systems. In those cases, we calculated volume through the morphometric relationships reported in Lewis (2011).

*Manning coefficient and channel form parameters*. These parameters were set at the values provided in section 2.1.3.

### 3.1.2 Human population and diclofenac consumption

*Population raster*. Human population was obtained from the Gridded Population of the World version 4 (GPWv4) (Doxsey-Whitfield et al., 2015). GLOBAL-FATE has been designed to overcome these constraints, offering the first contaminant fate model operating at the global river network, including lakes and reservoirs, which is at the same time open-source, multiplatform, user-friendly, and modular. This will make global contaminant calculations accessible to a much wider

community of scientists and practitioners, opening the door for including pharmaceutical pollution into influential assessments of climate change impacts (e.g., the Inter Sectoral Impact Model Intercomparison project) and global policy instruments like the UN Sustainable Development Goals agenda. GLOBAL-FATE calculates the steady-state concentration of a user-defined down-the-drain contaminant through the global river network, including lakes and reservoirs. GLOBAL-FATE is offered as an open-source, GIS-based model programmed in the C language, allowing researchers to select the input information (water routing, hydrology, population, etc.) and the spatial resolution at which the model has to perform. So forth, the model can include new or different hydrological datasets and other input information, and hence it is not fundamentally restricted to a single modelling resolution, hydrological, or socio-economic scenario. The model simulates the propagation of down-the-drain contaminants along the river network, and the constituent decreases at a rate proportional to its concentration in the aquatic media. GLOBAL-FATE is also computationally efficient, can be run in Windows or Linux machines, and can take advantage of parallel computing in multi-processor computers or clusters. It can also be run as a user-friendly plug-in in QGIS, and the modular structure of its code allows switching different functions of the model on and off.

*Per capita consumption raster*. The per capita consumption of diclofenac was calculated from information provided by the IMS-Health dataset for the period 2011-2013 (Acuña et al. 2015). The IMS-Health dataset includes national consumption of diclofenac for 86 nations (expressed as kilograms of consumed compound per year). Therefore, national consumption for the remaining 145 nations had to be estimated. Although IMS-Health data was only available for 38% of the global nations, these included the most populous and up to 82% of the global population. National per capita consumption for the 86 nations included in the IMS-Health dataset was estimated as the total consumption divided by the national population. The per capita consumption values of nations not included in the IMS-Health dataset were estimated as equal as the adjacent nation consumption (using Adjacent Fields function of ArcMap, ESRI; Acuña et al. (2015)).

*Excretion parameter*. We considered the oral application because it is the main form of administration and account for about 70% of the worldwide diclofenac sales following IMS-Health data (Zhang et al., 2008). We took $\gamma = 12.5\%$ as mean value for excretion rate (Johnson et al., 2013, $\gamma = 9.5\%$; Heberer and Feldmann, 2005, $\gamma = 10 - 15\%$; while Ternes et al., 1998, $\gamma = 15\%$).

### 3.1.3 WWTP and river removal

*Fraction of sewage treated raster*. Data of the fraction of wastewater that is treated per country were provided by the framework of "Environmental Performance Index" (EPI, Hsu et al., 2016) of the Yale University. Data were downloaded from https://epi.envirocenter.yale.edu/epi-downloads, and we produced a raster dataset with values per country.

*Fraction of contaminant attenuation in WWTP and first order decay rate in the river network*. The percentage of removal of diclofenac in water treatment plants, $\varepsilon = 40\%$, was decided as a tentative value between 21-40% and 69% (ranges from data in Zhang et al. (2008) and Ternes et al. (1998)). For this example, the first order decay rate in the river network was set to $k = 0.0096$ (after Pistocchi et al. 2012).

## 3.2 Model application and testing

Model predictions were obtained with a run-time of 5 minutes using a Desktop PC with Intel Core i5-4590 CPU 3.30 GHz and 8 GB RAM. The global concentration of diclofenac throughout the river network (Fig. 3) shows large areas of the World with very low concentration of diclofenac (mainly in boreal and tropical latitudes), while densely populated areas, particularly in Europe, Asia, and Africa show very high concentrations, sometimes beyond 100 ng L$^{-1}$. Thresholds of diclofenac concentrations for lowest observed effect on life concentration (LOEC, 30 ng L$^{-1}$) (Acuña et al. 2015) and the maximum acceptable limit proposed by the Water Framework Directive EC and the predicted non effect concentrations (PNEC) (both at 100 ng L$^{-1}$, Grill et al. 2016) are crossed in extensive regions of the World (Fig. 3). Simulated concentrations of diclofenac above 100 ng L$^{-1}$ are detected in isolated areas of North America, several areas in Central America and punctually in South America. In Africa concentrations over the above thresholds occur in the occidental Mediterranean coast (Fig. 3 and 4), Nigeria, and oriental and south-east sides of the continent. Furthermore, punctual areas in European countries show very high diclofenac concentrations, with remarkable prevalence in Belgium, central Europe and Ukraine (Fig. 4). Concentrations over the thresholds are also found in occidental Asia. India and Bangladesh stand out, mainly in the Ganges basin. Several regions of China, Thailand, and Japan also show very high concentrations. Concentrations above 100 ng L$^{-1}$ are also observed in some Indonesian islands, such as the Java Island.

The concentration maps in Figs. 3 and 4 do not show pixels with less than 100 mm year$^{-1}$ of runoff, which correspond to arid regions. We decided to discard concentration values in these areas because the quality of the runoff product we used is very poor below this threshold (Fekete et al., 2002), so any result would be unreliable. In addition, we also identified unrealistic, huge diclofenac concentrations in large urban areas due to unrealistic representation of river reaches and water infrastructure at our working resolution in these areas (sewage infrastructure in large urban areas is not accounted for in our model). To overcome this limitation, no diclofenac concentration is reported for cells accumulating contaminant mass for less than three upstream cells i.e., $l < 3$ in Eq. (5). The two filters described above exemplify how the interpretation of GLOBAL-FATE outcomes depends on the available input datasets, both in terms of quality and resolution. Considering that working resolution and input datasets are user-dependent in GLOBAL-FATE, the criteria to assess model results quality and reliability are case dependent, and the filters suggested here may not be convenient in all circumstances. In any case, users must be aware that the simplified representation of complex processes like water and contaminant routing along natural and

engineered systems currently coded in GLOBAL-FATE implies serious limitations on the spatial scale at which the model delivers meaningful results (see Section 4 for a comprehensive discussion on this issue).

Although the aim of this exercise was only to exemplify the application of GLOBAL-FATE in a real case, we assessed the goodness-of-fit of model predictions against observed loads of diclofenac in the river network. We used 405 diclofenac loading (concentration times streamflow) values measured in rivers around the globe compiled by Acuña et al. 2015, and compared this with the modelled value in the corresponding cell after log-transforming the two values (the range of observed and modelled diclofenac loadings shows several orders of magnitude). We used the Nash–Sutcliffe model efficiency coefficient to assess model performance:

$$E = 1 - \frac{\sum \left( L_{obs} - L_{est} \right)^2}{\sum \left( L_{obs} - \overline{\left( \log L_{obs} \right)} \right)^2} \tag{19}$$

The relationship between observed and simulated diclofenac loads (Fig. 5) shows a Nash–Sutcliffe model efficiency of 0.4, which is reasonable considering that we did not calibrate any parameter of the model. Global models for contaminants always suffer from low to medium performance scores due to the scarce and spatially biased datasets available for model evaluation (Strokal et al., 2019), and frequently they only go beyond E>0.5 after intensive calibration procedures (e.g., Harrison et al, 2019).

### 3.3 Sensitivity analysis

Model simulations may diverge from observed values due to uncertainty in observations and parametric values, and to deliberate simplifications inherent in all phases of the modelling process. Furthermore, most input datasets come from previous modelling exercises with more assumptions and simplifications that may affect the final result. We carried out a sensitivity analysis in order to investigate the propagation of errors to the output from a selection of inputs (population, pharmaceutical consumption, excretion rate, runoff, decay rate in WWTPs and the river network, lakes volume, Manning coefficient, and the $d$ exponent in Eq. 11). This analysis was performed using a local sensitivity, one-at-a-time procedure, changing one input per simulation around a reference parametric point, defined by the values of the original datasets or the parametric value provided in section 3.1. These inputs were perturbed around this reference point, decreasing and increasing the value from -100% to 100% the original figure in 10% increments. In case of raster datasets, the whole domain was perturbed in a homogeneous way. We assessed the sensitivity of the mean diclofenac load in the river network to those perturbations in the inputs, expressed as percent change form the value in the reference condition.

The results from the sensitivity analysis (Fig. 6) suggest that the output of the model is highly sensitive to two groups of inputs. On one hand, everything related to the generation of contaminant mass by population (population, consumption, and excretion rate) showed the largest overall sensitivity (the sensitivities are the same for these parameters because they are multiplying themselves in the model, Eq. 14). On the other hand, the output was also very sensitive to parameters related to the attenuation of contaminant in the river network: the first order decay rate in the river network, and parameters related to water residence time calculation such as the Manning coefficient and the exponent $d$ for water depth. The output showed less sensitivity to the rest of tested inputs. These results suggest that the quality of datasets related to the generation of contaminant from human use must be carefully checked, and that attenuation of the contaminants in rivers and lakes plays an important role on defining their presence in the river network. This last point is very relevant considering that data for first order reaction rates in rivers for many contaminants are scarce or non-existent, and that residence time calculation in the river network still depends on global empirical functions that may have large regional variability. Also, these results suggest that mitigation strategies to reduce the prevalence of pharmaceutical contaminants in the river network should point to increasing the assimilation of the drug by the human body and decisions and campaigns devoted to lower the per capita consumption. This would be much more efficient than increasing WWTP treatment technologies to attenuate the contaminant load before reaching the river network, at least in regions where the prevalence of wastewater treatment is already high.

## 4. Strengths and limitations of GLOBAL-FATE

GLOBAL-FATE is an open-source, multi-platform, and modular contaminant fate model that links human consumption of pharmaceutical-like compounds with their concentration in the river network. GLOBAL-FATE is also computationally efficient, and can solve the whole global streamflow generation and contaminant routing in less than five minutes in a customary PC. It provides practical guidelines (through readme files and example datasets) to assist non-specialist users in computer programming. At the same time, it has a fully commented code that experienced users can easily customize and further develop to adapt to their needs. The model is also available as a user-friendly QGIS plug-in. Through simple menus, an inexperienced user can conduct simulations and produce basic outputs on the QGIS canvas. This will make global contaminant calculations accessible to a much wider community of scientists and practitioners.

One of the features of GLOBAL-FATE is that it is not fundamentally associated with a spatial resolution or extent. Users can define the working spatial resolution and extent just adapting the resolution of the raster inputs and the region of interest (for instance, a single continent or subcontinent). Although this is an obvious advantage over other large-scale contaminant models, it also harbors the significant risk that users may assume that the model delivers meaningful results at any working spatial scale. We strongly advise against the uncritical use of GLOBAL-FATE, particularly when working at coarse working resolutions or with highly spatially aggregated input data. We do not want to suggest a spatial resolution threshold from which results from GLOBAL-FATE could be considered as reliable, because the criteria to assess model results quality and

reliability are case dependent, and guidelines suggested in a given situation may not be convenient in all circumstances. In our example, the combination of the working spatial scale (1/16 of a degree, ~7 km), the complexity of fine-scale interactions between engineered systems and the river network (e.g., the exact location of effluent discharges, extensive sewage networks, poor representation of small streams), and the input data available translates into several model inadequacies that pose limits on the interpretation of the results. We already mentioned that the quality of the runoff map precludes the interpretation of any results for regions where runoff is below 100 mm year$^{-1}$, and that the calculated concentration are unreliable for watersheds smaller than ~150 km$^2$ (this roughly relates to river reaches of ~20 km) due to inexact effluent discharge locations in small streams and the absence of data on sewage networks in large urban areas, that would route the contaminant load downstream towards larger rivers resulting in higher dilution and lower contaminant concentration. These limitations were easily spotted as they resulted in very unrealistic high diclofenac concentrations scattered throughout the global network, which attracted our immediate attention. However, other assumptions of the modelling approach do not leave such a conspicuous mark in the model output. For instance, consumption data is homogeneous at the country level, while variability inside large countries may be substantial (urban vs. rural regions, for instance). Also, we have averaged information on intensity of treatment also at the country scale, when this may change even at very local scales. This implies that the model results are not necessarily unbiased beyond the threshold mentioned earlier (150 km$^2$), because all uncertainties and biases propagating from model inputs and assumptions must also have a reflection in the spatial dimension at varying scales. For instance, the comparison between observed and modelled diclofenac concentration along the main axis of the Rhine river (Fig. 7) shows that the model was able to spot a concentration increase at 300 km upstream the river mouth (in the sense that the model predicts an increase that goes beyond 100 ng L$^{-1}$, the basic threshold we were interested in). However, in the same basin close to the river mouth (∼50 km) the model could not mimic an increase in concentration beyond 100 ng L$^{-1}$. Our opinion is that GLOBAL-FATE, as implemented in the example, should be used to answer questions which are general in nature. For instance, "contaminant concentration downstream large urban areas in Central Europe frequently exceeds 100 ng L$^{-1}$", and related statements concerning remediation measures. We advise against the use of GLOBAL-FATE as implemented in the example to support statements concerning particular places at or near the working resolution (for instance "the remediation measures seem insufficient to lower concentrations below 100 ng L$^{-1}$ downstream from Cologne"). We acknowledge that this restriction limits the usability of the current version of GLOBAL-FATE to answer questions that require precise information at a scale of ~20 km of river network. In such cases, models operating at local (i.e., single watershed scale) or regional (i.e., country level) scales may be a better option (e.g., Diamantini et al., 2019). In any case, GLOBAL-FATE can be used to test the effectiveness of large scale management strategies related to pharmaceutical consumption control and wastewater treatment implementation and upgrading, in order to deliver influential assessments of climate change impacts on pharmaceutical consumption and river network ecosystem health (e.g., the Inter Sectoral Impact Model Intercomparison project), and also for informing global policy instruments like the UN Sustainable Development Goals agenda. This is already common practice in other sectors using large scale, coarse resolution models such as impacts of climate change on marine life (Lotze et al., 2019), on lake physics (Woolway and Merchant

2019), on soil moisture (Samaniego et al., 2018), or on economic losses due to river flooding (Dottori et al., 2018), to cite just a few recent examples.

Nonetheless, we discussed the limitations of GLOBAL-FATE as applied in our example (~7 km pixel resolution), but even exercises using models working at much finer resolution in smaller areas (e.g., China at 0.5 km resolution, Grill et al. 2018) found substantial uncertainties related to unaccounted variability regarding input variables and poor representation of small streams. Therefore, we strongly suggest to carefully assess model performance irrespective of the working resolution, and to pay special attention to the spatial scales at which answers are required and its compatibility with the aggregation of input information and the representation of the river network. Finally, GLOBAL-FATE includes very simplified physically based approximations for attenuation in the river network, which a priori are mathematically robust to changes in the spatial resolution, but that assumes homogeneous properties along calculation units (river reaches) such as water velocity and mixing. Although those assumptions do not hold even at very local scales (tens of meters), empirical research on river ecology suggests that this approach is reasonable for rivers reaches up to ~10 km (Marcé et al., 2018). Beyond this, substantial heterogeneity of the river network is overlooked, with potential effects on the contaminant mass balance (Darracq and Destouni, 2007).

GLOBAL-FATE is a steady state model, and although synoptic conditions like low o high flows or climate change scenarios can be modelled, it cannot dynamically simulate extreme events or seasonality. This should be considered when formulating research or management questions for which hydrological seasonal or subseasonal variability is relevant. A keen aspect of GLOBAL-FATE is that researchers are free to use the input information they prefer, it is not limited to particular hydrological products, so synoptic conditions can always be modelled, as far as the steady state assumption is reasonable.

GLOBAL-FATE is the first contaminant model operating at the global scale that fully integrates lakes and reservoirs in the routing of a contaminant along the river network. This is a relevant improvement over other modelling approaches, especially considering the long water residence time of lakes and reservoirs compared to river reaches, which implies a prominent role of lakes and reservoirs on the attenuation of contaminants. However, it should be noted that GLOBAL-FATE models lakes and reservoirs as point-like features, with no spatial heterogeneity. This may fail to capture likely gradients of contaminant concentration in large lakes and reservoirs.

Our analyses showed that GLOBAL-FATE will have a performance in terms of goodness-of-fit similar to other global contaminant fate models. However, as in any other modelling exercise, this will be highly dependent on the quality of input data used and the availability of observed data to adjust parameters that cannot be set at confident values using prior information. In any case, large uncertainties will always be present in global models including simplified lumped

representations of very complex processes. We pointed to the main limitations of the model and the most sensitive inputs, but researchers will have to re-assess this in a case-by-case fashion.

As for future developments, we envisage the inclusion of diffuse pollution in the current steady-state framework, which would make GLOBAL-FATE useful for a much wider range of pollutants, such as nutrients or agricultural pesticides, and a

500 more detailed accounting of sewage infrastructure to be able to solve contaminant routing at high resolution in very large urban areas. In any case, GLOBAL-FATE will be a valuable tool for the scientific community and the policymaking arena, and could be used to test the effectiveness of large scale management strategies related to pharmaceutical consumption control and wastewater treatment implementation and upgrading.

**Code availability**

The GLOBAL-FATE code (including compiling instructions, examples, and the QGIS plug-in) is available in the following URL: https://github.com/ICRA/GLOBALFATE. Prebuilt executables for Windows are available under request.

**Author contribution**

RM, VA, and SS conceived the model. RM produced a preliminary version of the model code, later modified and expanded by FB, CF, and VA. CF and FB built the diclofenac example and performed model runs. CF and RM prepared the manuscript

with contributions from all co-authors.

**Acknowledgements**

This work has been supported by the European Communities 7th Framework Programme funding under Grant Agreement No.603629-ENV-2013-6.2.1-GLOBAQUA. Authors also acknowledge the support for scientific equipment given by the European Regional Development Fund (FEDER) under the Catalan FEDER Operative Program 2007-2013 and by MINECO

according to DA3ª of the Catalan Statute of Autonomy and to PGE-2010.

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

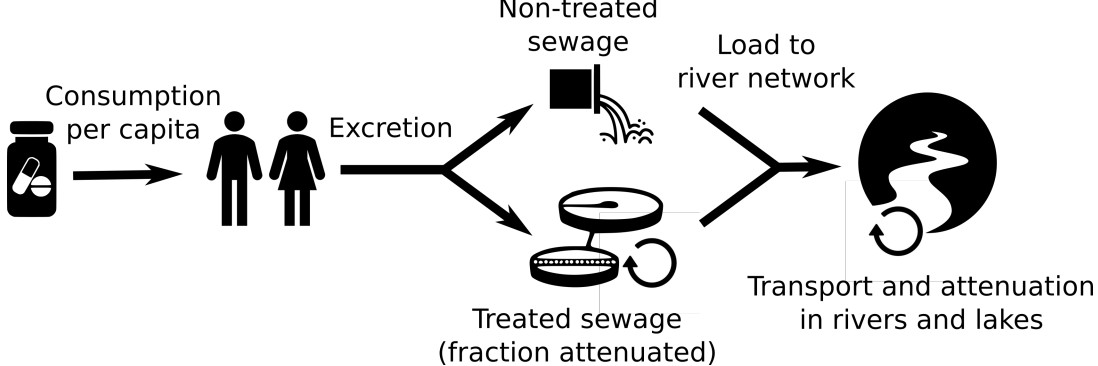

**Figure 1. Conceptual diagram of the processes modelled by GLOBAL-FATE**

**Figure 2. Work flow of GLOBAL-FATE**


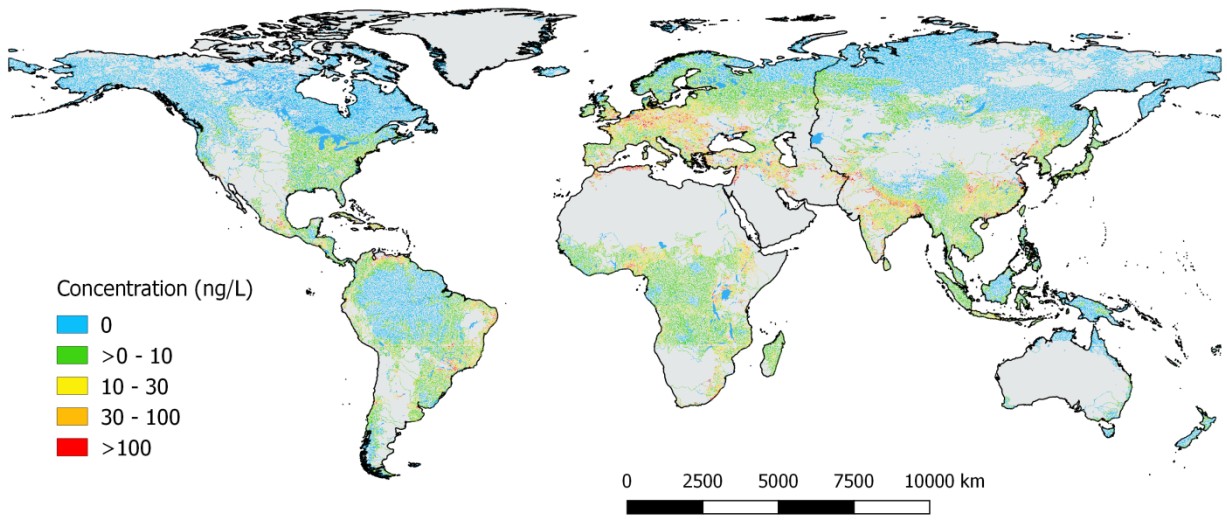


**Figure 3. Simulated mean annual diclofenac concentration worldwide.**



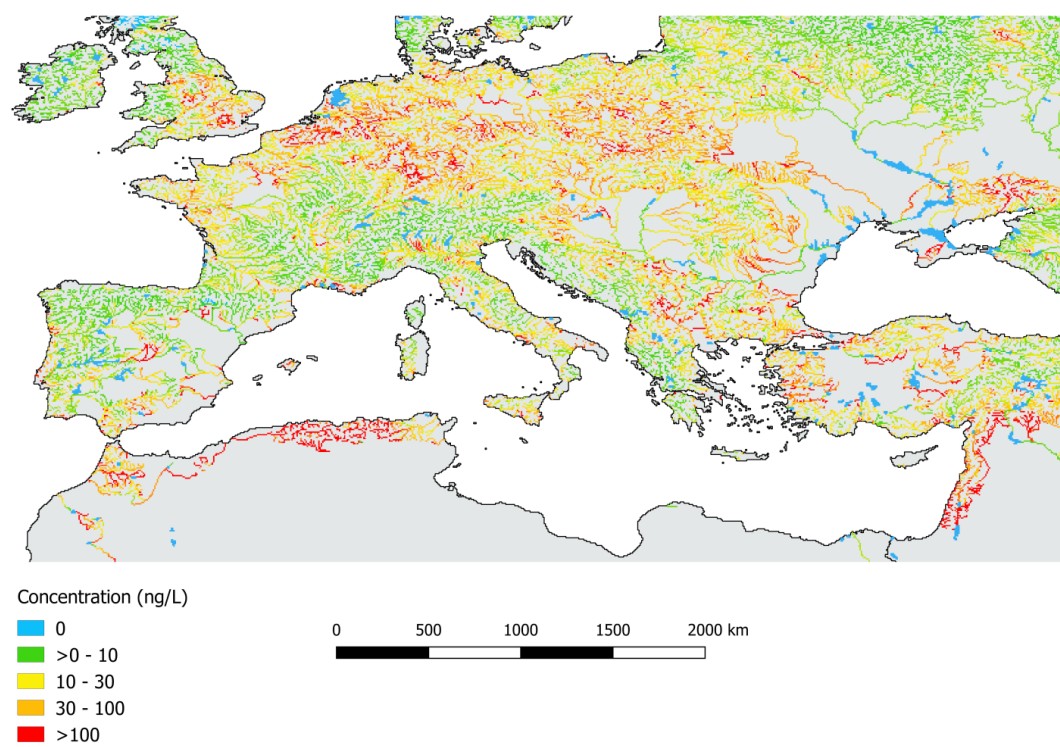

**Figure 4. Simulated mean annual diclofenac concentration in Central and Southern Europe and the southern Mediterranean basin.**



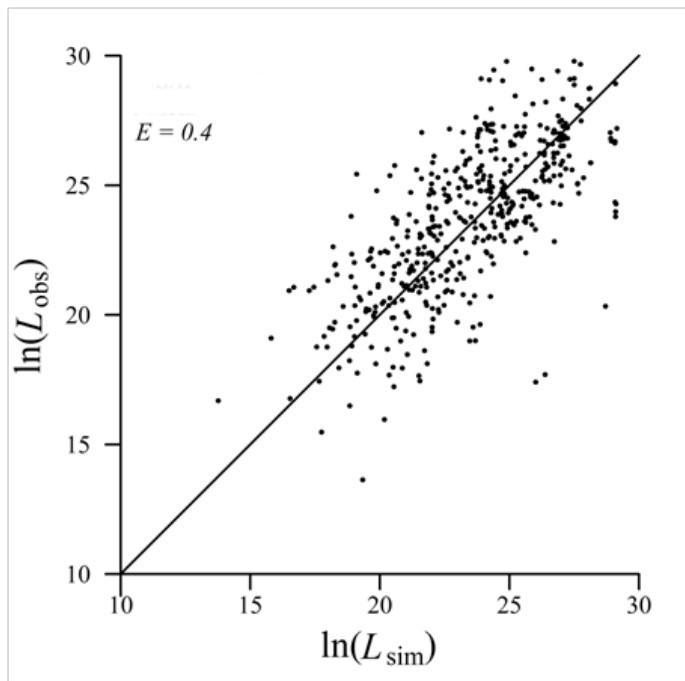

**Figure 5. Observed versus simulated load log-values (** $\ln(ng/L)$ **),** $N = 405$ **points. Nash–Sutcliffe model efficiency coefficient (E) is also reported.**

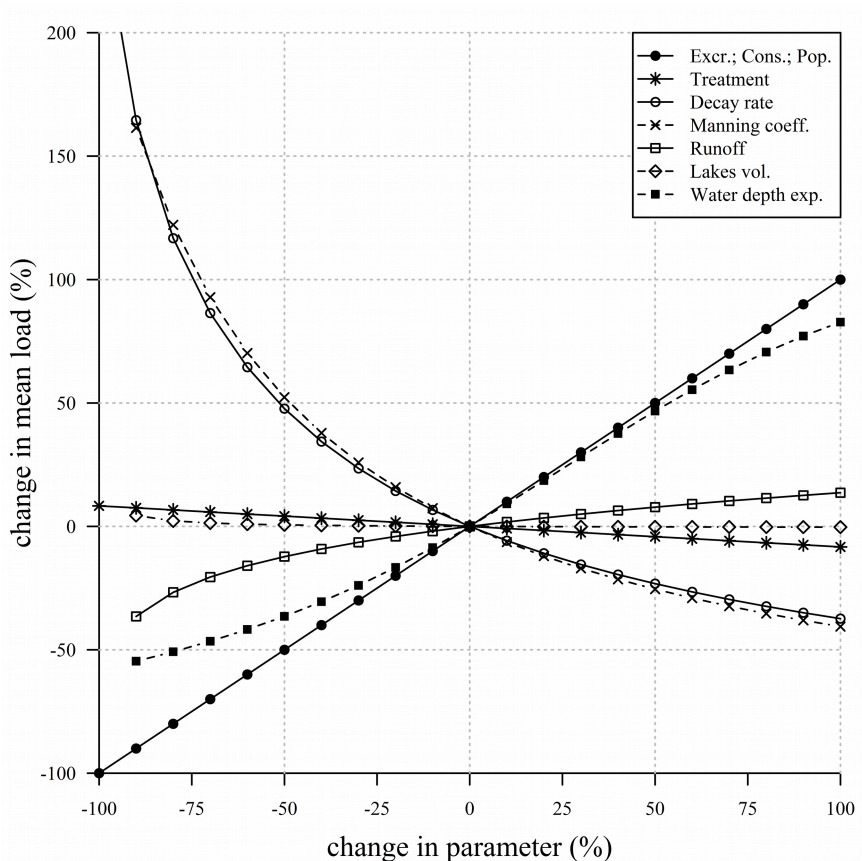

**Figure 6. Spider plot of percent changes in the mean load in the river network due to changes in a collection of inputs to GLOBAL-FATE.**


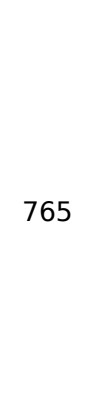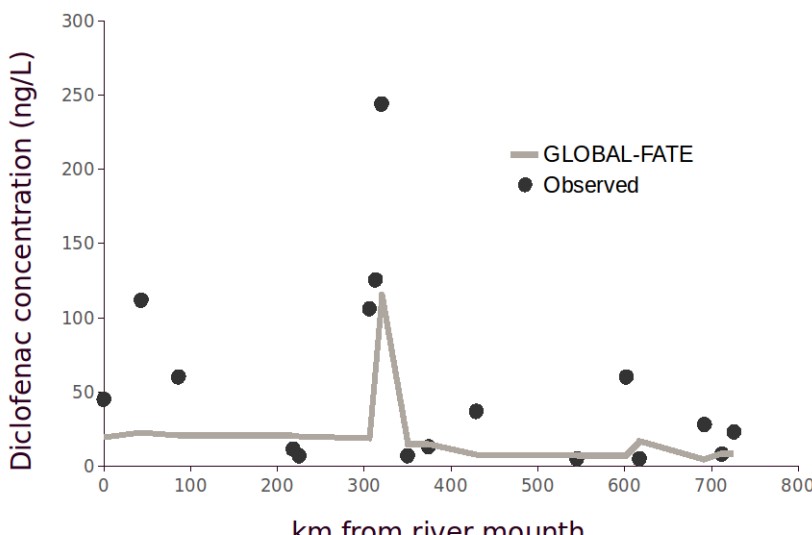

**Figure 7. GLOBAL-FATE diclofenac simulation along the Rhine river (Europe), and diclofenac observations from our compiled database.**

| | PhATE | GREAT-ER | LF2000-WQX | HydroROUT | iSTREEM | ePiE | GLOBAL-FATE |
|---|---|---|---|---|---|---|---|
| **Pollutant modelled** | Pharmaceuticals | Diclofenac and propranolol; nonylphenol and nonylphenol ethoxylates; Sulfamethoxazole (antibiotic) | Pharmaceuticals Steroid estrogens | Pharmaceuticals | HHCB and DEET; triclosan and carbamazepine | Pharmaceuticals | Pharmaceuticals |
| **Spatial extent** | US | Large river basins | England and Walles | Quebec and Ontario | US | EU | World |
| **Spatial resolution** | Discrete segments (~16 km) | Discrete segments | River reach visible at a scale of 1:50,000 | Raster of 500 m pixel resolution | River network segment | 30 arc seconds (~1 km) | Any raster resolution |
| **Model type** | Deterministic | Mixes deterministic with stochastic processes | Mixed deterministic and stochastic model | Deterministic | Deterministic | Deterministic | Deterministic |
| **Sources of pollutants** | Point sources, from Publicly owned treatment works (POTW) | Point sources from WWTP | Point sources | Point sources | Point sources from WWTP. Different treatments in WWTP | Point sources from WWTP | Point sources |
| **Model implementation and availability** | Microsoft Visual C++ and uses Microsoft Access databases | Implemented as part of a GIS. It is open source software under the GNU Public License | Not available | Not available | Public web application | Written in R | Public code written in C. QGIS plug-in |
| **Transferability to global scale** | Limited geographic scope | Restricted to river network dataset and WWTP information availability | Restricted to river network dataset and WWTP information availability | It needs other models (WaterGAP) to estimate runoff | Restricted to river network dataset and WWTP information availability | Restricted to river network dataset and WWTP information availability | Native |
| **References** | Anderson et al., 2004 | Johnson et al. 2007; Zhang et al., 2015; Archundia et al., 2018 | Boxall et al., 2014; Keller et al., 2015 | Grill et al., 2016 | Kapo et al., 2015; Ferrer and DeLeo, 2017 | Oldenkamp et al., 2018 | This study |


**Table 1. Features of a collection of contaminants fate models compared to those of GLOBAL-FATE.**

|  | INPUTS | OUTPUTS |
|---|---|---|
| **Morphology and Hydrology** | • Flow direction<br>• Area accumulation (hierarchic structure)<br>• Runoff (mm year$^{-1}$)<br>• Lakes location and shape<br>• Lakes volume (m$^3$)<br>• Slope (m m$^{-1}$)<br><br>○ Manning coefficient (s m$^{-1/3}$)<br>○ Parameters for channel form (4 of them) | • Cells area (m$^2$) and width (m)<br>• Streamflow (m$^3$ year$^{-1}$)<br>• Residence time in rivers and lakes (hours)<br>• Lake outlet discharge (m$^3$ year$^{-1}$) |
| **Contaminant** | • Population (people per cell)<br>• Contaminant consumption per capita (country level, g person$^{-1}$ year$^{-1}$)<br>• Population connected to WWTPs (country level, fraction)<br><br>○ Decay constant in the river network (hour$^{-1}$)<br>○ Human excretion rate (fraction)<br>○ WWTP attenuation efficiency (fraction) | • Contaminant concentration (g m$^{-3}$) |

**Table 2. Input and output datasets and parameters for both geographical (morphology and hydrology) and contaminant model processes. Filled bullets represent raster datasets, non-filled bullets stand for parameters.**

| Process | Description | Inputs | Outputs | C function |
|---|---|---|---|---|
| Area | Calculates cells area | ❖ No direct user inputs, but projection must be WGS84 | • Area for each cell in latitude direction* ($m^2$) <br> • Horizontal cells width for each cell in latitude direction* (m) | Area_m2 _fun.c |
| Flow routing | Calculates streamflow | ❖ Raster of flow direction <br> ❖ Raster of area accumulation <br> ❖ Raster of runoff (m year$^{-1}$) <br> ➢ Area ($m^2$) | • Raster of streamflow* ($m^3$ year$^{-1}$) | Flow_accumula tion_m2.c |
| Residence Time calculator | Calculates residence time for every cell | ❖ Raster of slope (m m$^{-1}$) <br> ❖ Manning coefficient (s m$^{-1/3}$) <br> ❖ Parameters of channel form (4) <br> ➢ Raster of streamflow ($m^3$ year$^{-1}$) <br> ➢ Cell height and cell width (m) <br> ✓ Raster of flow direction <br> ✓ Raster of area accumulation | • Raster of flow velocity* (m s$^{-1}$) <br> • Raster of residence time in rivers (hours) | RT_rivers_calc ulator.c |
| Lakes RT incorporation | Incorporates lakes into the RT raster | ❖ Raster of lakes location and shape <br> ❖ Raster of lakes volume ($m^3$) | • Raster of residence time in rivers and lakes* (hours) <br> • Vector of outlet discharge per lake* ($m^3$ year$^{-1}$) | RT_lakes_incor poration.c |
| Contaminant load | Calculates consumption by population and attenuation in WTTPs | ❖ Population raster (people per pixel) <br> ❖ Raster of pharmaceutical consumption per capita (g person$^{-1}$ year$^{-1}$) <br> ❖ Raster of fraction of sewage treated <br> ❖ Rate of contaminant excretion <br> ❖ Rate of contaminant removal in WWTP | • Raster of contaminant load from human consumption to the river network (g year$^{-1}$) | Initial_conta minant_load. c |
| Contaminant routing | Calculates contaminant routing in the river network | ❖ Exponential decay rate (hours $^{-1}$) <br> ➢ Raster of residence time (hours) <br> ➢ Raster of streamflow ($m^3$ year$^{-1}$) <br> ✓ Raster of flow direction <br> ✓ Raster of area accumulation | • Raster of contaminant concentration* (g m$^{-3}$) or load* (g year$^{-1}$) in the river network | Contaminant_a ccumulation.c |

**Input flags legend:**
❖  Dataset used for the first time
➢  Input coming from previous functions output
✓  Data set used (at least) for the second time

**Table 3. Main calculation steps in GLOBAL-FATE, with indication of inputs and outputs used by each process, and the C functions responsible. Outputs with an asterisks can be saved during model execution and accessed afterwards.**