# Peer review of "GLOBAL-FATE: A GIS-based model for assessing contaminants fate in the global river network"

_Geoscientific Model Development, 2019_

## Short Comment (SC1) · 14 Mar 2019

Dear authors,

In my role as Executive editor of GMD, I would like to bring to your attention our Editorial version 1.1:

http://www.geosci-model-dev.net/8/3487/2015/gmd-8-3487-2015.html

This highlights some requirements of papers published in GMD, which is also available on the GMD website in the 'Manuscript Types' section:

http://www.geoscientific-model-development.net/submission/manuscript_types.html

In particular, please note that for your paper, the following requirements have not been

met in the Discussions paper:

- "The main paper must give the model name and version number (or other unique identifier) in the title."

- "If the model development relates to a single model then the model name and the version number must be included in the title of the paper. If the main intention of an article is to make a general (i.e. model independent) statement about the usefulness of a new development, but the usefulness is shown with the help of one specific model, the model name and version number must be stated in the title. The title could have a form such as, "Title outlining amazing generic advance: a case study with Model XXX (version Y)"."

- "All papers must include a section, at the end of the paper, entitled 'Code availability'. Here, either instructions for obtaining the code, or the reasons why the code is not available should be clearly stated. It is preferred for the code to be uploaded as a supplement or to be made available at a data repository with an associated DOI (digital object identifier) for the exact model version described in the paper. Alternatively, for established models, there may be an existing means of accessing the code through a particular system. In this case, there must exist a means of permanently accessing the precise model version described in the paper. In some cases, authors may prefer to put models on their own website, or to act as a point of contact for obtaining the code. Given the impermanence of websites and email addresses, this is not encouraged, and authors should consider improving the availability with a more permanent arrangement. After the paper is accepted the model archive should be updated to include a link to the GMD paper."

Note, that the exact code version described in this article should be permanently accessible. Thus please consider to make the exact version, your article refers to, available

via a permanent archive providing a DOI (e.g. Zenodo). Additionally, please add a version number identifying this version to the title of your article upon submission of the revised manuscript.

Yours,

Astrid Kerkweg

---

## Referee Comment (RC1) · Alberto Bellin (Referee) · 29 Apr 2019

This manuscript presents a generalization of the FATE model for global applications. The model inherits all the simplifying assumptions and limitations of FATE and focuses on providing a GIS platform suitable for global applications.

General comments

The modeling part is simplistic, as in FATE, and boils down to the application of the following first order decay equation, providing the contribution of a cell to the annual load observed at the reference cell: $L = L_0 exp[-k\tau]$, where $L$ is the contaminant annual load $[g/year]$ at the reference cell, $L_0$ is proportional to the population of the contributing cell and $\tau$ is the residence time from the contributing cell to the reference

one. The loads of all the cells contributing to the reference one are added such as to obtain the total load, which is then divided by the annual water discharge at the same cell. This approach does not capture important mechanisms, such as seasonality in the releases, temperature and hydrology, which may cause significant fluctuations of the contaminant concentration. This is somewhat acknowledged by the authors in the discussion.

Recently, we published a more comprehensive model (Diamantini et al., 2019), which under suitable assumptions can be reduced to the approach presented in the manuscript, but that is more general and allows to take into account the above processes. This previous published work includes also the effect of lakes that the authors claim they introduced for the first time. I think that the work we did is relevant to this contribution since it represents a generalization of the proposed approach.

I appreciated the disclaimer the authors introduced in the conclusions, where they warned users against the application of the model at what they call the "very local" scale. However this scale is not adequately defined, though by mentioning the watershed scale as an example of scale at which the model cannot be applied and the following suggestion of not using the model "below the country level" provides some, but still ambiguous, guidelines. This notwithstanding, the disclaimer poses strong limitations to the analyses that can be done and a more comprehensive discussion about the limits of applications is needed, in my view, to avoid misuses of the proposed model. Considering that the model cannot provide valuable information at important scales, such as the watershed scale and downstream large urban areas (see sentence beginning at line 9 of page 15), where the impacts are evaluated, I am wondering what type of indications the model can actually provide, besides suggesting the reduction of drug consumption, a recommendation that can be done by considering the total consumption based on census information. In other words, my concern is that hydrological processes may not be so relevant for the type of questions that the model can actually answer, considering the level of simplification introduced, thereby making this model

not clearly preferable to alternative approaches, such as simple regressions or machine learning, for example. A discussion supporting the utility of the model is needed here.

The authors remark that GLOBAL-FATE is not associated to a spatial resolution, or extent, and consider this as the "main strength" of the proposed approach. I disagree with this conclusion. The size of the cell has an impact on the way the river systems are represented and a coarse gridding may produce inaccurate estimates of the residence time. For instance, the raster of 1/16 degree used in the example of application is already too coarse and does not guarantee a good reproduction of the river system in densely populated areas, such as in Europe for example. On the other hand, this gridding may be ok in large rivers with low population density, but as a consequence with low impact. An upper limit should be indicated here and a warning to avoid improper applications with large cells should be issued.

Detailed comment

I am wondering how the value that the NS assumes after log-transforming the data compares with that obtained without the transformation. In Figure 5 the points are rather disperse and this may be due to the attenuating effect of errors when the log-transform is applied.

Alberto Bellin

References

Diamantini, E., Mallucci, S., Bellin, A. 2019 A parsimonious transport model of emerging contaminants at the river network scale. Hydrol. Earth Syst. Sci., 23(1): 573-593, doi:10.5194/hess-23-573-2019.

---

## Author Comment (AC1) · 26 Jun 2019

We appreciate the thoughtful revision by Dr. Alberto Bellin, and want to make some precisions about our modelling strategy and additional clarifications that we think addresses the reviewer concerns.

First, we want to make clear that GLOBAL-FATE is not a revision or upgrade of the PhATE model, as the first comment by the reviewer seems to suggest. Although some assumptions and approaches are shared between GLOBAL-FATE and PhATE, and in fact with a number of other contaminant models, the development of our model has been totally independent of the former or any other contaminant model (except of course for the inspiration and guidance collected from all past work on large scale

modelling we found in the literature).

We totally agree that our modelling approach is simple, but we do disagree with the adjective "simplistic". We decided to work with such a simple model structure because we understand that working at global scales precludes any attempt to parameterize a complex model including a lot of processes, such as relationship with temperature or seasonality of releases. This is particularly true if the available information collected in the field is scarce, as it is the case. A limitation to the level of detail captured in GLOBAL-FATE is the need of consistent and complete datasets with global coverage and the variety of sources and unknown sampling methodologies that make difficult to use the data as reference datasets.

It is a piece of fundamental knowledge that any attenuation process in the river network depends on temperature one way or the other, but there are simply not enough data in the literature to parameterize this dependence at large scales. And the seasonality of releases, as modelled by GLOBAL-FATE, would need global, gridded information on seasonality of population. To the best of our knowledge, such a data product does not exist yet. All in all, our choices concerning model structure were not the result of a naive approach to the problem (and thus "simplistic"), but of a careful consideration of pros and cons considering the kind of questions we are anticipating to answer with a model like this and the information available to parameterize a model at such large scales.

There is no doubt that hydrology exerts a prominent role in defining pollutant concentrations in the river network, including seasonal variations. In fact, we already make this point clear in section 4 in the manuscript, where we discuss limitations of GLOBAL-FATE. However, working at annual, average streamflow conditions does not preclude the model being useful to answer many relevant questions concerning pollution in the river network. It precludes, for instance, to answer questions about the impacts of extreme events, although there is the possibility to run the model for different synoptic situations. We do not deny that seasonal and short-term hydrological variability is relevant for contaminant transport, we simply do not intent to answer questions related to this variability with GLOBAL-FATE. Needless to say, the decision to work with annual averaged streamflow assuming steady-state was also related to the complexity and computing needs that come with a dynamical hydrological model working at daily or hourly time steps.

The model introduced by Diamantini et al. (2019) is a very fine work, and we agree that constitutes a basic antecedent that we must incorporate into the final version of our manuscript. It nicely incorporates time-varying forcing functions like population and hydrology. And, indeed, it links lakes and reservoirs with the river network. However, the fact that the authors simulated a small-medium watershed (12.000 km2) to exemplify the applicability of their model already shows which are the spatial scales for which this model has been conceptualized. In our opinion and making clear that we do not diminish in any way Diamantini and co-workers' approach, that we like very much, it is not fair comparing the complexity in terms of model structure of models devised to work at such diverging spatial and temporal scales. On one hand, we posit that applying Diamantini's model at the global scale using the same approach as in the original paper would imply a gigantic (and most probably unsuccessful) effort related to parameterization of the model and computing resources. On the other hand, we acknowledge that applying GLOBAL-FATE to answer the questions posed by Diamantini and co-workers in the Adige basin would be inappropriate.

We thank Dr. Bellin for his insightful comments on the spatial scales at which GLOBAL-FATE deliver meaningful and usable results. We agree that we were not particularly brilliant at this respect, as we introduced some ambiguity and vagueness that did not help to convey the message. This point arises from the fact that GLOBAL-FATE does not have pre-defined working scales, and the users can decide which data products to use to force the simulations. We already stated in the paper that GLOBAL-FATE outcomes depends on the available datasets, both in terms of quality and resolution. Considering that working resolution and input datasets are user-dependent in GLOBAL-FATE, the

criteria to assess model results quality and reliability are case dependent, and guide-lines suggested in a given situation may not be convenient in all circumstances. There-fore, with the aim to be as conservative as possible, we stated in the discussion that: "we doubt that GLOBAL-FATE in its current formulation can be used for simulations at very local scales, like single watersheds". We were thinking in very small watersheds in the upstream sections of the river network, but we clearly added a lot of confusion here. We also stated that "Although this would depend on each model implementation, we suggest not using results from GLOBAL-FATE to draw conclusions below the country level". Clearly, this was too conservative, because we used GLOBAL-FATE to suggest patterns below the country level in the paper, and we are already drawing conclusions below the country level in non-yet-published applications of the tool. We included this statement to prevent users to draw conclusions at very local spatial scales (pixel) from GLOBAL-FATE applications using coarse resolutions (since the working resolution is user defined) and scarce supporting information, but again, in absence of a clear dis-cussion of this topic this added even more confusion.

Considering this, we will rework the sections of the paper addressing this issue, to make clear that GLOBAL-FATE is not attached to a resolution nor to a data product, and because of this it is difficult to provide guidelines about working scales that would apply to any potential application. Therefore, a careful analysis of the outcomes and the implications of the spatial scale chosen is always a must. However, we understand the point of the referee when it comes to the example provided in the paper, and the re-vised version of the manuscript will include more precise indication of the spatial scales at which conclusions can be drawn. In particular, we will include an analysis showing the implications of working at the single-pixel scale in our example, consisting mainly in unrealistic very high concentrations in pixels draining small (1 or 2 pixels) watersheds in large urban areas (that would be connected to waterworks serving several pixels in most occasions). This effect greatly decreases when removing from the analysis all one-pixel and two-pixel watersheds, suggesting that the model results are unreliable when assessed in areas smaller than approx. 150km2. This roughly related to rivers

reaches of approx. 20km. We interpret this scale as the fundamental limitation of the GLOBAL-FATE model as applied in our example, clearly indicated by the available empirical information However, we also have to consider that beyond the limitation related to large urban areas and the lack of detail on water infrastructures, the application of GLOBAL-FATE in the examples includes other simplifications that do not leave such a conspicuous mark in the model output. For instance, consumption data is homogeneous at the country level, while variability inside large countries can be large (urban vs. rural, for instance). Also, we have averaged information on intensity of treatment also at the country scale, when this may change even at very local scales. Considering this, we do not want to oversell the former conclusion that GLOBAL-FATE can deliver meaningful results at a scale of 150km2 resolution. Although our analysis supports that the model delivers acceptable simulations at this scale, the comparison between observed and modelled values clearly indicates that there is high uncertainty that must also have a reflection in the spatial dimension.

For instance, we have an example of the limitations of our tool as implemented in the paper. This is the observed contaminant concentration along the main axis of the Rhine river (Fig. 1 in this revision). We can see that the model was able to spot a concentration increase at 300 km upstream the river mouth (in the sense that the model predicts an increase that goes beyond 100 ng/L, the basic thershold we were interested in). However, in the same basin close to the river mouth (∼50 km) the model did not mimetize an increase in concentration beyond 100 ng/L.

FIGURE 1 HERE

Therefore, our opinion is that GLOBAL-FATE, as implemented in the example, should be used to answer questions which are general in nature. For instance, "contaminant concentration downstream large urban areas in Central Europe frequently exceeds 100 ng/L", and related statements concerning remediation measures, for instance. We think that statements concerning particular places at or near the working resolution should be avoided (for instance "the remediation measures seem insufficient to lower concentrations below 100 ng/L downstream from Cologne"). However, we agree with the referee that the scale-free feature of GLOBAL-FATE is at the same time a strength and a limitation. However, to our understanding this limitation is not related to a fundamental flaw in the structure of GLOBAL-FATE, but just to the possibility that users may want to answer a given question using a spatial resolution and data products that could be inappropriate for that particular purpose, delivering misleading results and conclusions. Although this reasoning would apply to any modeling exercise, we understand the point of Dr. Bellin that claiming that GLOBAL-FATE is not tied to a particular resolution may promote bad modelling practices. Consequently, we will issue a warning in the paper to avoid improper applications using too coarse cells or poor/scarce information. In any case, we want to add at this point that the freedom that GLOBAL-FATE provides to choose the spatial scale also works in the other direction, that is, the user is free to work with a much finer resolution than the one used in the example.

Concerning the comment about the log scale used for comparing observed and modelled values, we had no other option considering that the magnitude of the errors was proportional to the modelled value. This effect in a modelling exercise spanning 3 orders of magnitude forced us to use the log scale for a proper calibration of the tool. Additionally, we spotted a typo in equation 19 that will be fixed.

We thank Dr. Bellin again for the nice revision, and we are happy to continue the discussion on the different aspects of the revision.

Carme Font and co-authors.
* * *
[Figure]

**Fig. 1.** Example of simulation along the Rhine river

---

## Referee Comment (RC2) · Bethanna Jackson (Referee) · 3 Sep 2019

Although this manuscript is much improved compared to an earlier submission to this journal, it still is not clearly demonstrating a contribution to new ideas /methods; I would like to see in a revision, significant effort on further positioning it versus other models and methods to demonstrate a uniqueness of GLOBAL-FATE versus other models and software available.

I am in strong agreement with the other referee, that the current main argument does not fully hold - it is not appropriate to consider that GLOBAL-FATE is not associ- ated to a spatial resolution. It may RUN at any spatial resolution, but that is very different to it being methodologically appropriate at any spatial resolution. Not only

detailed physics, but even very simplified physically based approximations to processes/integration of rates of change understanding break down once space and/or time steps become too large. Please add very strong warnings about upper limits. So what new thing/contribution is being brought? Is it a science contribution, a software contribution that allows others as well as you to take the science further, which is still a contribution, or both?

I also asked a colleague without a specific understanding of contaminant transport, but with a strong computational modelling background, to do a useability review, which I provide below: as per the comments on science and code, note its not damning but not yet convinced if its great worth ...

*********************************************** A technical review of

GLOBAL-FATE: A GIS-based model for assessing contaminants fate in the global river network by Carme Font et al. (2019)

Main executable

I was able to get the GLOBAL-FATE executable running on my PC and tested it with the sample data provided. It took approximately 20 minutes to run, but my laptop processor (Intel Core i5-2435M CPU @ 2.40GHz) is not as powerful as the one mentioned in the paper, so this running time seems about right compared with the five minutes given in the paper.

It would be useful to provide Windows binaries in the repository rather than users having to compile them themselves. As I was not familiar with this process this took me quite some time. If you would prefer not to supply the binaries, then some clearer instructions would be helpful, especially with regards to installing Cygwin.

While there are some comments within the C code, these could be improved and added to, to allow users such as myself to gain a better understanding of what the code is doing.

QGIS plugin

I also tried to use the GLOBAL-FATE QGIS plugin. It would be helpful if you mention that the plugin is only compatible with QGIS 2 and not QGIS 3 as I first installed QGIS 3.6 in order to try this plugin but was told that I had to use QGIS 2 when I tried to install the plugin (I used QGIS 2.18).

It would also be useful to provide some brief instructions on how to install the QGIS plugin for non-expert users such as myself.

When running the plugin I came across the error that the directory 'C:\tmp' had not been created (IOError: [Errno 2] No such file or directory: 'C:/tmp/dir.txt'). This error could be mitigated by either creating the directory for the user if it does not exist, or by asking the user for a temporary directory as one of the inputs. I simply created the directory as a workaround to this problem.

The input parameters are split into two dialog boxes. Could these be combined into one dialog box as this could be more intuitive to the user?

When the GLOBAL-FATE plugin started executing, another dialog box popped up immediately giving the elapsed time. I was unable to determine why the GLOBAL-FATE code was not executing, so I was unable to run the plugin successfully. Also, I was unsure where the data would be saved to, or if maps showing the data would just load within QGIS.

---

## Author Response (AR1)

**Response to Referee Bethanna Jackson**

**REFEREE: Although this manuscript is much improved compared to an earlier submission to this journal, it still is not clearly demonstrating a contribution to new ideas /methods; I would like to see in a revision, significant effort on further positioning it versus other models and methods to demonstrate a uniqueness of GLOBAL-FATE versus other models and software available.**

AUTHORS: We thank Dr. Jackson for this comment, which is somehow in agreement with a concern raised by Alberto Bellin. In the new version of the manuscript, we put particular emphasis on this aspect, from the abstract to the discussion. We understand GLOBAL-FATE as an advanced in the state-of-the-art of global contaminant modelling, offering in a single tool features that are scattered in several models, but not available in a single application. Therefore, we do not consider GLOBAL-FATE as offering novel scientific advances in the way we parametrize the processes at play, but it would undoubtedly make global contaminant modelling accessible to a much wider community of scientist and policymakers. This will ultimately help the progress of large-scale contaminant modelling offering an open source and flexible platform to test new parameterizations (hypothesis), and also allowing policymakers to plan global or continental strategic actions.

We have modified the manuscript to make this point much clearer, including statement in several places:

- Line 11-Abstract : "GLOBAL-FATE is the first open-source, multiplatform, user-friendly, and modular contaminant fate model operating at the global scale linking human consumption of pharmaceutical-like compounds with their concentration in the river network."

- Line 23-Abstract: "GLOBAL-FATE will be a valuable tool for the scientific community and the policymaking arena, and could be used to test the effectiveness of large scale management strategies related to pharmaceutical consumption control and wastewater treatment implementation and upgrading."

- Line 62: "GLOBAL-FATE has been designed to overcome these constraints, offering the first contaminant fate model operating at the global river network, including lakes and reservoirs, which is at the same time open-source, multiplatform, user-friendly, and modular. This will make global contaminant calculations accessible to a much wider community of scientists and practitioners, opening the door for including pharmaceutical pollution into influential assessments of climate change impacts (e.g., the Inter Sectoral Impact Model Intercomparison project) and global policy instruments like the UN Sustainable Development Goals agenda. GLOBAL-FATE calculates the steady-state concentration of a user-defined down-the-drain contaminant through the global river network, including lakes and reservoirs. GLOBAL-FATE is offered as an open-source, GIS-based model programmed in the C language, allowing researchers to select the input information (water routing, hydrology, population, etc.) and the spatial resolution at which the model has to perform. So forth, the model can include new or different hydrological datasets and other input information, and hence it is not fundamentally restricted to a single modelling resolution, hydrological, or socio-economic scenario. The model simulates the propagation of down-the-drain contaminants along the river network, and the constituent decreases at a rate proportional to its concentration in the aquatic media. GLOBAL-FATE is also

computationally efficient, can be run in Windows or Linux machines, and can take advantage of parallel computing in multi-processor computers or clusters. It can also be run as a user-friendly plug-in in QGIS, and the modular structure of its code allows switching different functions of the model on and off."

Line 83: "GLOBAL-FATE is a physically-based model for simulating constituent inputs to the river network and their routing along the river network at the global scale. Our approach shares key assumptions and modelling mechanisms with other large scale pharmaceutical models for the river network (i.e., Keller et al. 2006; Pistocchi 2014; Grill et al., 2019), including the use of per capita mass emissions of the contaminant of interest, simplified parameterization of losses due to human metabolism and removal in wastewater treatment plants, and dilution and first order attenuation dynamics upon discharge into natural waters. However, GLOBAL-FATE is the first model natively operating at the global scale including all those mechanisms, including explicit routing and attenuation in lakes and reservoirs."

- Line 418: "GLOBAL-FATE is an open-source, multiplatform, and modular contaminant fate model that links human consumption of pharmaceutical-like compounds with their concentration in the river network. GLOBAL-FATE is also computationally efficient, and can solve the whole global streamflow generation and contaminant routing in less than five minutes in a customary PC. It provides practical guidelines (through readme files and example datasets) to assist non-specialist users in computer programming. At the same time, it has a fully commented code that experienced users can easily customize and further develop to adapt to their needs. The model is also available as a user-friendly QGIS plug-in. Through simple menus, an inexperienced user can conduct simulations and produce basic outputs on the QGIS canvas. This will make global contaminant calculations accessible to a much wider community of scientists and practitioners."

**I am in strong agreement with the other referee, that the current main argument does not fully hold - it is not appropriate to consider that GLOBAL-FATE is not associated to a spatial resolution. It may RUN at any spatial resolution, but that is very different to it being methodologically appropriate at any spatial resolution. Not only detailed physics, but even very simplified physically based approximations to processes/integration of rates of change understanding break down once space and/or time steps become too large. Please add very strong warnings about upper limits.**

We cannot but agree with Dr. Jackson at this point, which was also raised by Dr. Bellin. We acknowledge that we somehow oversold the scale-free feature of GLOBAL-FATE, because although it is a potential advantage over other available models, it also leaves the door open for gross misuses of the model. To avoid this, we worked in two directions: first, we substantially expanded the section where we assess the limitations of GLOBAL-FATE as implemented in the example, including a new figure showing detailed results for a single watershed; and second, we included a clear warning in the discussion about the use of GLOBAL-FATE at low resolutions.

For the first point, we included the following text (lines 363-375):

[revised manuscript text omitted]

Also, we totally agree that physically-based formulations may lose its physical meaning when working at resolutions very far from the ones used to conceptualize the underlying model. Therefore, we included the following lines in the paper, lines 474-480:

"Finally, GLOBAL-FATE includes very simplified physically based approximations for attenuation in the river network, which a pripori are mathamatically robust to changes in the spatial resolution, but that assumes homogenous propierties along calculation units (river reaches) such as water velocity and mixing. Although those assumptions do not hold even at very local scales (tens of meters), empirial reearch on river ecology suggests that this approach is reasonable for rivers reaches up to ~10 km (Marcé et al., 2018). Beyond this, substantial hetereogenity of the river network is overlooked, with potential effects on the contaminant mass balance (Darracq and Destouni, 2007)."

**So what new thing/contribution is being brought? Is it a science contribution, a software contribution that allows others as well as you to take the science further, which is still a contribution, or both?**
Please, see the response to the first comment that already addressed this concern.

**I also asked a colleague without a specific understanding of contaminant transport, but with a strong computational modelling background, to do a usability review, which I provide below: as per the comments on science and code, note its not damning but not yet convinced if its great worth**

We thank the reviewer for this detailed technical assessment, and we want to apologize because we upload the wrong version of the plug-in code into GitHub, which was the ultimate reason of the problems encountered when trying to use it, as detailed below.

**A technical review of GLOBAL-FATE: A GIS-based model for assessing contaminants fate in the global river network by Carme Font et al. (2019)**

**Main executable**
**I was able to get the GLOBAL-FATE executable running on my PC and tested it with the sample data provided. It took approximately 20 minutes to run, but my laptop processor (Intel Core i5-2435M CPU @ 2.40GHz) is not as powerful as the one mentioned in the paper, so this running time seems about right compared with the five minutes given in the paper.**

**It would be useful to provide Windows binaries in the repository rather than users having to compile them themselves. As I was not familiar with this process this took me quite some time. If you would prefer not to supply the binaries, then some clearer instructions would be helpful, especially with regards to installing Cygwin.**

Regarding the installation of the model, we are not allowed to load executable files in Github, but we have included clear indication that executables will be sent to any user under request, both in the main body of the paper (under the section Code availability) , and also in the instructions at the GitHub site.

**While there are some comments within the C code, these could be improved and added to, to allow users such as myself to gain a better understanding of what the code is doing.**

We have included a lot more comments in the code. It would be cumbersome to detail here everything we added, but we invite the referee to consult the source code files to have an idea of the commenting level of the new version, which we think it is high.

**QGIS plugin**

**I also tried to use the GLOBAL-FATE QGIS plugin. It would be helpful if you mention that the plugin is only compatible with QGIS 2 and not QGIS 3 as I first installed QGIS 3.6 in order to try this plugin but was told that I had to use QGIS 2 when I tried to install the plugin (I used QGIS 2.18). It would also be useful to provide some brief instructions on how to install the QGIS plugin for non-expert users such as myself.**

The first version of the plugin was made for QGIS 2, now we have been working on a new version for QGIS 3. Therefore, the final version of the plug-in will work on QGIS 3. For the installation of the plugin, we updated the file in Github explaining all the steps to follow:
https://github.com/icra/GLOBALFATE/blob/master/QGIS%20plug-in/INSTALL.txt

**When running the plugin I came across the error that the directory 'C:\tmp' had not been created (IOError: [Errno 2] No such file or directory: 'C:/tmp/dir.txt'). This error could be mitigated by either creating the directory for the user if it does not exist, or by asking the user for a temporary directory as one of the inputs. I simply created the directory as a workaround to this problem.**

We apologize because we also encountered this error during plug-in development, and it was already solved. However, for some unknown reason, the version of the plug-in we uploaded to Github still included the bug, which we see precluded a proper assessment of the tool. As the referee suggested, the workaround consisted in creating creating the directory directly from the python script, so the user wouldn't have to bother on it.

**The input parameters are split into two dialog boxes. Could these be combined into one dialog box as this could be more intuitive to the user?**

The reason of two dialog boxes is that you can switch off some modules of the code (for instance, the hydrological calculations), for instance to run different scenarios. This is why we decided to split the input in two boxes. The second box only appears if hydrological calculations are performed. We think that this is better than having a single window, because it may confound users that will input those files even in cases when they are not necessary.

**When the GLOBAL-FATE plugin started executing, another dialog box popped up immediately giving the elapsed time. I was unable to determine why the GLOBAL-FATE code was not executing, so I was unable to run the plugin successfully. Also, I was unsure where the data would be saved to, or if maps showing the data would just load within QGIS.**

We apologize again because we also encountered this error during plug-in development, and it was already solved. However, for some unknown reason, the version of the plug-in we uploaded to Github also included this bug, which we see precluded a proper assessment of the tool. The point of this error was the presence of an absolute address that precludes the execution of the model in different computers. This bug is now solved, with a box in the main menu of the plugin, where the user can specify the directory in which to save the results. We apologize the referee could not assess the tool

because this error. Just for the record, the main result, i.e. the map of contaminants concentrations, is automatically loaded in the canvas.

**Response to Referee Alberto Bellin**

**REFEREE: This manuscript presents a generalization of the FATE model for global applications. The model inherits all the simplifying assumptions and limitations of FATE and focuses on providing a GIS platform suitable for global applications.**

AUTHORS: We appreciate the thoughtful revision by Dr. Alberto Bellin, and want to make some precisions about our modelling strategy and additional clarifications that we think addresses the reviewer concerns. First, we want to make clear that GLOBAL-FATE is not a revision or upgrade of the PhATE model, as the first comment by the reviewer seems to suggest. Although some assumptions and approaches are shared between GLOBAL-FATE and PhATE, and in fact with a number of other contaminant models, the development of our model has been totally independent of the former or any other contaminant model (except of course for the inspiration and guidance collected from all past work on large scale modelling we found in the literature).

**General comments**
**The modeling part is simplistic, as in FATE, and boils down to the application of the following first order decay equation, providing the contribution of a cell to the annual load observed at the reference cell: $L = L_0 \exp[-k\tau]$, where $L$ is the contaminant annual load [g/year] at the reference cell, $L_0$ is proportional to the population of the contributing cell and $\tau$ is the residence time from the contributing cell to the reference one. The loads of all the cells contributing to the reference one are added such as to obtain the total load, which is then divided by the annual water discharge at the same cell. This approach does not capture important mechanisms, such as seasonality in the releases, temperature and hydrology, which may cause significant fluctuations of the contaminant concentration. This is somewhat acknowledged by the authors in the discussion.**

We totally agree that our modelling approach is simple, but we do disagree with the adjective "simplistic". We decided to work with such a simple model structure because we understand that working at global scales precludes any attempt to parameterize a complex model including a lot of processes, such as relationship with temperature or seasonality of releases. This is particularly true if the available information collected in the field is scarce, as it is the case. A limitation to the level of detail captured in GLOBAL-FATE is the need of consistent and complete datasets with global coverage and the variety of sources and unknown sampling methodologies that make difficult to use the data as reference datasets.

It is a piece of fundamental knowledge that any attenuation process in the river network depends on temperature one way or the other, but there are simply not enough data in the literature to parameterize this dependence at large scales. And the seasonality of releases, as modelled by GLOBAL-FATE, would need global, gridded information on seasonality of population. To the best of our knowledge, such a data product does not exist yet. All in all, our choices concerning model structure were not the result of a naive approach to the problem (and thus "simplistic"), but of a careful consideration of pros

and cons considering the kind of questions we are anticipating to answer with a model like this and the information available to parameterize a model at such large scales.

There is no doubt that hydrology exerts a prominent role in defining pollutant concentrations in the river network, including seasonal variations. In fact, we already make this point clear in section 4 in the manuscript, where we discuss limitations of GLOBAL-FATE. However, working at annual, average streamflow conditions does not preclude the model being useful to answer many relevant questions concerning pollution in the river network. It precludes, for instance, to answer questions about the impacts of extreme events, although there is the possibility to run the model for different synoptic situations. We do not deny that seasonal and short-term hydrological variability is relevant for contaminant transport, we simply do not intent to answer questions related to this variability with GLOBAL-FATE. Needless to say, the decision to work with annual averaged streamflow assuming steady-state was also related to the complexity and computing needs that come with a dynamical hydrological model working at daily or hourly time steps.

However, we understand the referee's concerns, and have included a number of warnings and considerations in the manuscript. The main modifications are listed here:

- We made clear that the degree of simplification in GLOBAL-FATE is similar to other large-scale contaminant models, such as for instance the most recent development in Grill, G., Li, J., Khan, U., Zhong, Y., Lehner, B., Nicell, J., Ariwi, J.: Estimating the eco-toxicological risk of estrogens in China's rivers using a high-resolution contaminant fate model, Water Research, 145, 707-720, https://doi.org/10.1016/j.watres.2018.08.053, 2018.

- We expanded the discussion on the limitations of our modelling approach due to the rough temporal resolution for hydrology. Particularly, we have in line 466 these sentences:

"GLOBAL-FATE is a steady state model, and although synoptic conditions like low o high flows or climate change scenarios can be modelled, it cannot dynamically simulate extreme events or seasonality. This should be considered when formulating research or management questions for which hydrological seasonal or subseasonal variability is relevant."

**Recently, we published a more comprehensive model (Diamantini et al., 2019), which under suitable assumptions can be reduced to the approach presented in the manuscript, but that is more general and allows to take into account the above processes. This previous published work includes also the effect of lakes that the authors claim they introduced for the first time. I think that the work we did is relevant to this contribution since it represents a generalization of the proposed approach.**

The model introduced by Diamantini et al. (2019) is a very fine work, and we agree that constitutes a basic antecedent that we have incorporated in our discussion about the limitations of our approach. It nicely incorporates time-varying forcing functions like population and hydrology. And, indeed, it links lakes and reservoirs with the river network. However, the fact that the authors simulated a small-medium watershed (12.000 km$^2$) to exemplify the applicability of their model already shows which are the spatial scales for which this model has been conceptualized. In our opinion and making clear that

we do not diminish in any way Diamantini and co-workers' approach, that we like very much, it is not fair comparing the complexity in terms of model structure of models devised to work at such diverging spatial and temporal scales. On one hand, we posit that applying Diamantini's model at the global scale using the same approach as in the original paper would imply a gigantic (and most probably unsuccessful) effort related to parameterization of the model and computing resources. On the other hand, we acknowledge that applying GLOBAL-FATE to answer the questions posed by Diamantini and co-workers in the Adige basin would be inappropriate. We tried to incorporate this reasoning in the new version of the manuscript in several places, but we think this new sentence catches the point quite conveniently (line 443):

"We acknowledge that this restriction limits the usability of the current version of GLOBAL-FATE to answer questions that require precise information at a scale of ~20 km of river network. In such cases, models operating at local (i.e., single watershed scale) or regional (i.e., country level) scales may be a better option (e.g., Diamantini et al., 2019)."

**I appreciated the disclaimer the authors introduced in the conclusions, where they warned users against the application of the model at what they call the "very local" scale. However this scale is not adequately defined, though by mentioning the watershed scale as an example of scale at which the model cannot be applied and the following suggestion of not using the model "below the country level" provides some, but still ambiguous, guidelines. This notwithstanding, the disclaimer poses strong limitations to the analyses that can be done and a more comprehensive discussion about the limits of applications is needed, in my view, to avoid misuses of the proposed model. Considering that the model cannot provide valuable information at important scales, such as the watershed scale and downstream large urban areas (see sentence beginning at line 9 of page 15), where the impacts are evaluated, I am wondering what type of indications the model can actually provide, besides suggesting the reduction of drug consumption, a recommendation that can be done by considering the total consumption based on census information. In other words, my concern is that hydrological processes may not be so relevant for the type of questions that the model can actually answer, considering the level of simplification introduced, thereby making this model not clearly preferable to alternative approaches, such as simple regressions or machine learning, for example. A discussion supporting the utility of the model is needed here.**

We thank Dr. Bellin for his insightful comments on the spatial scales at which GLOBAL- FATE deliver meaningful and usable results, and the implications for the overall value of our model. We agree that we were not particularly brilliant at this respect, as we introduced some ambiguity and vagueness that did not help to convey the message. In fact, this point also puzzled the Associate Editor, who also wondered what was the main contribution of our approach.

In the new version, we have tried to locate our model in the landscape of contaminant models in a more explicit way, making a strong case for the step forward GLOBAL-FATE constitutes. First, making clear which is the technical novelty GLOBAL-FATE offers (lines 55-76):

"Recently, other approaches specifically designed for very large scales have used a Geographical Information System (GIS) framework to solve the routing of chemicals along the river network (Pistocchi et al., 2012; Dumont et al., 2015; Grill et al., 2016; Rice and Westerhoff 2017). Most of these models use a much simpler model parameterization, in order to make continental and global

calculations accessible. However, some of them assume that chemicals do not decay when travelling through the river network, and simply rely on dilution factors once pollutants enter in the river network. Further, they work at a fixed spatial scale which is either very rough to adequately represent the river network (e.g., 0.5 degrees), or too detailed to be practical for global calculations due to computational requirements (e.g., 500 m, Grill et al., 2018).

GLOBAL-FATE has been designed to overcome these constraints, offering the first contaminant fate model operating at the global river network, including lakes and reservoirs, which is at the same time open-source, multiplatform, user-friendly, and modular. This will make global contaminant calculations accessible to a much wider community of scientists and practitioners, opening the door for including pharmaceutical pollution into influential assessments of climate change impacts (e.g., the Inter Sectoral Impact Model Intercomparison project) and global policy instruments like the UN Sustainable Development Goals agenda. GLOBAL-FATE calculates the steady-state concentration of a user-defined down-the-drain contaminant through the global river network, including lakes and reservoirs. GLOBAL-FATE is offered as an open-source, GIS-based model programmed in the C language, allowing researchers to select the input information (water routing, hydrology, population, etc.) and the spatial resolution at which the model has to perform. So forth, the model can include new or different hydrological datasets and other input information, and hence it is not fundamentally restricted to a single modelling resolution, hydrological, or socio-economic scenario. The model simulates the propagation of down-the-drain contaminants along the river network, and the constituent decreases at a rate proportional to its concentration in the aquatic media. GLOBAL-FATE is also computationally efficient, can be run in Windows or Linux machines, and can take advantage of parallel computing in multi-processor computers or clusters. It can also be run as a user-friendly plug-in in QGIS, and the modular structure of its code allows switching different functions of the model on and off. "

Also, from line 461 on, we state that:

"In any case, GLOBAL-FATE can be used to test the effectiveness of large scale management strategies related to pharmaceutical consumption control and wastewater treatment implementation and upgrading, in order to deliver influential assessments of climate change impacts on pharmaceutical consumption and river network ecosystem health (e.g., the Inter Sectoral Impact Model Intercomparison project), and also for informing global policy instruments like the UN Sustainable Development Goals agenda. This is already common practice in other sectors using large scale, coarse resolution models such as impacts of climate change on marine life (Lotze et al., 2019), on lake physics (Woolway and Merchant 2019), on soil moisture (Samaniego et al., 2018), or on economic losses due to river flooding (Dottori et al., 2018), to cite just a few recent examples."

**The authors remark that GLOBAL-FATE is not associated to a spatial resolution, or extent, and consider this as the "main strength" of the proposed approach. I disagree with this conclusion. The size of the cell has an impact on the way the river systems are represented and a coarse gridding may produce inaccurate estimates of the residence time. For instance, the raster of 1/16 degree used in the example of application is already too coarse and does not guarantee a good reproduction of the river system in densely populated areas, such as in Europe for example. On the other hand, this gridding may be ok in large rivers with low population density, but as a consequence with low impact. An upper limit should be indicated here and a warning to avoid improper applications with large cells should be issued.**

We acknowledge that we somehow oversold the scale-free feature of GLOBAL-FATE, because although it is a potential advantage over other available models, it also leaves the door open for gross misuses of the model. To avoid this, we worked in two directions: first, we substantially expanded the section where we assess the limitations of GLOBAL-FATE as implemented in the example, including a new figure showing detailed results for a single watershed; and second, we included a clear warning in the discussion about the use of GLOBAL-FATE at low resolutions.

For the first point, we included the following text (lines 363-375):

[revised manuscript text omitted]

**Detailed comment**

**I am wondering how the value that the NS assumes after log-transforming the data compares with that obtained without the transformation. In Figure 5 the points are rather disperse and this may be due to the attenuating effect of errors when the log-transform is applied.**

Concerning the comment about the log scale used for comparing observed and modelled values, we had no other option considering that the magnitude of the errors was proportional to the modelled value. This effect in a modelling exercise spanning 3 orders of magnitude forced us to use the log scale for a proper calibration of the tool.

**Response to Executive editor Astrid Kerkweg**

**COMMENT: In my role as Executive editor of GMD, I would like to bring to your attention our Editorial version 1.1: http://www.geosci-model-dev.net/8/3487/2015/gmd-8-3487-2015.html. This highlights some requirements of papers published in GMD, which is also available on the GMD website in the 'Manuscript Types' section: http://www.geoscientific-model-development.net/submission/manuscript_types.html. In particular, please note that for your paper, the following requirements have not been met in the Discussions paper:**

**15• "The main paper must give the model name and version number (or other unique identifier) in the title."**
**• "If the model development relates to a single model then the model name and the version number must be included in the title of the paper. If the main intention of an article is to make a general (i.e. model independent) statement about the usefulness of a new development, but the usefulness is shown with the help of one specific model, the model name and version number must be stated in the title. The title could have a form such as, "Title outlining amazing generic advance: a case study with Model XXX (version Y)"."**

AUTHORS: We now refer to the model as GLOBAL-FATE (version 1.0.0) in the paper and the Github repository.

**• "All papers must include a section, at the end of the paper, entitled 'Code availability'. Here, either instructions for obtaining the code, or the reasons why the code is not available should be clearly stated. It is preferred for the code to be uploaded as a supplement or to be made available at a data repository with an associated DOI (digital object identifier) for the exact model version described in the paper. Alternatively, for established models, there may be an existing means of accessing the code through a particular system. In this case, there must exist a means of permanently accessing the precise model version described in the paper. In some cases, authors may prefer to put models on their own website, or to act as a point of contact for obtaining the code. Given the impermanence of websites and email addresses, this is not encouraged, and authors should consider improving the availability with a more permanent arrangement. After the paper is accepted the model archive should be updated to include a link to the GMD paper." Note, that the exact code version described in this article should be permanently accessible. Thus please consider to make the exact version, your article refers to, available via a permanent archive providing a DOI (e.g. Zenodo). Additionally, please add a version number identifying this version to the title of your article upon submission of the revised manuscript.**

AUTHORS: We are storing the exact version of the code in a Github repository. We are ready to link this to Zenodo, but we are still waiting for permission of the owner of the ICRA Github repository to make an explicit link of the GLOBAL-FATE repository to Zenodo. After this step, which is going to be completed by early October 2019, we will be able to provide a DOI for the GLOBAL-FATE (version 1.0.0) repository.

[revised manuscript text omitted]

• Area accumulation (hierarchic structure)
• Runoff (mm year$^{-1}$)
• Lakes location and shape
• Lakes volume (m$^3$)
• Slope (m m$^{-1}$)

○ Manning coefficient (s m$^{-1/3}$)
○ Parameters for channel form (4 of them) | • Cells area (m$^2$) and width (m)
• Streamflow (m$^3$ year$^{-1}$)
• Residence time in rivers and lakes (hours)
• Lake outlet discharge (m$^3$ year$^{-1}$) |
| **Contaminant** | • Population (people per cell)
• Contaminant consumption per capita (country level, g person$^{-1}$ year$^{-1}$)
• Population connected to WWTPs (country level, fraction)

○ Decay constant in the river network (hour$^{-1}$)
○ Human excretion rate (fraction)
○ WWTP attenuation efficiency (fraction) | • Contaminant concentration (g m$^{-3}$) |

780 **Table 2. Input and output datasets and parameters for both geographical (morphology and hydrology) and contaminant model processes. Filled bullets represent raster datasets, non-filled bullets stand for parameters.**

| Process | Description | Inputs | Outputs | C function |
|---|---|---|---|---|
| Area | Calculates cells area | ❖ No direct user inputs, but projection must be WGS84 | • Area for each cell in latitude direction* ($m^2$)
• Horizontal cells width for each cell in latitude direction* (m) | Area_m2 _fun.c |
| Flow routing | Calculates streamflow | ❖ Raster of flow direction
❖ Raster of area accumulation
❖ Raster of runoff (m year$^{-1}$)
➢ Area ($m^2$) | • Raster of streamflow* ($m^3$ year$^{-1}$) | Flow_accumulation_m2.c |
| Residence Time calculator | Calculates residence time for every cell | ❖ Raster of slope (m m$^{-1}$)
❖ Manning coefficient (s m$^{-1/3}$)
❖ Parameters of channel form (4)
➢ Raster of streamflow ($m^3$ year$^{-1}$)
➢ Cell height and cell width (m)
✓ Raster of flow direction
✓ Raster of area accumulation | • Raster of flow velocity* (m s$^{-1}$)
• Raster of residence time in rivers (hours) | RT_rivers_calculator.c |
| Lakes RT incorporation | Incorporates lakes into the RT raster | ❖ Raster of lakes location and shape
❖ Raster of lakes volume ($m^3$) | • Raster of residence time in rivers and lakes* (hours)
• Vector of outlet discharge per lake* ($m^3$ year$^{-1}$) | RT_lakes_incorporation.c |
| Contaminant load | Calculates consumption by population and attenuation in WTTPs | ❖ Population raster (people per pixel)
❖ Raster of pharmaceutical consumption per capita (g person$^{-1}$ year$^{-1}$)
❖ Raster of fraction of sewage treated
❖ Rate of contaminant excretion
❖ Rate of contaminant removal in WWTP | • Raster of contaminant load from human consumption to the river network (g year$^{-1}$) | Initial_contaminant_load.c |
| Contaminant routing | Calculates contaminant routing in the river network | ❖ Exponential decay rate (hours$^{-1}$)
➢ Raster of residence time (hours)
➢ Raster of streamflow ($m^3$ year$^{-1}$)
✓ Raster of flow direction
✓ Raster of area accumulation | • Raster of contaminant concentration* (g m$^{-3}$) or load* (g year$^{-1}$) in the river network | Contaminant_accumulation.c |

**Input flags legend:**
❖ Dataset used for the first time
➢ Input coming from previous functions output
✓ Data set used (at least) for the second time

785 **Table 3. Main calculation steps in GLOBAL-FATE, with indication of inputs and outputs used by each process, and the C functions responsible. Outputs with an asterisks can be saved during model execution and accessed afterwards.**